# High-resolution volumetric imaging constrains compartmental models to explore synaptic integration and temporal processing by cochlear nucleus globular bushy cells

George A Spirou[1]*[†], Matthew Kersting[1‡], Sean Carr[1], Bayan Razzaq[2§], Carolyna Yamamoto Alves Pinto[1#], Mariah Dawson[2¶], Mark H Ellisman[3,4], Paul B Manis[5,6]*[†]

[1]Department of Medical Engineering, University of South Florida, Tampa, United States; [2]Department of Otolaryngology, Head and Neck Surgery, West Virginia University, Morgantown, United States; [3]Department of Neurosciences, University of California, San Diego, San Diego, United States; [4]National Center for Microscopy and Imaging Research, University of California, San Diego, San Diego, United States; [5]Department of Otolaryngology/Head and Neck Surgery, University of North Carolina at Chapel Hill, Chapel Hill, United States; [6]Department of Cell Biology and Physiology, University of North Carolina, Chapel Hill, United States

*For correspondence: gspirou@usf.edu (GAS); pmanis@med.unc.edu (PBM)

[†]These authors contributed equally to this work

Present address: [‡]Noblis Inc, Reston, United States; [§]Department of Neurosurgery, Stony Brook University, Stony Brook, United States; [#]Department of Biomedical Engineering, Johns Hopkins University, Baltimore, United States; [¶]PNC Financial Services, Cleveland, United States

**Abstract** Globular bushy cells (GBCs) of the cochlear nucleus play central roles in the temporal processing of sound. Despite investigation over many decades, fundamental questions remain about their dendrite structure, afferent innervation, and integration of synaptic inputs. Here, we use volume electron microscopy (EM) of the mouse cochlear nucleus to construct synaptic maps that precisely specify convergence ratios and synaptic weights for auditory nerve innervation and accurate surface areas of all postsynaptic compartments. Detailed biophysically based compartmental models can help develop hypotheses regarding how GBCs integrate inputs to yield their recorded responses to sound. We established a pipeline to export a precise reconstruction of auditory nerve axons and their endbulb terminals together with high-resolution dendrite, soma, and axon reconstructions into biophysically detailed compartmental models that could be activated by a standard cochlear transduction model. With these constraints, the models predict auditory nerve input profiles whereby all endbulbs onto a GBC are subthreshold (coincidence detection mode), or one or two inputs are suprathreshold (mixed mode). The models also predict the relative importance of dendrite geometry, soma size, and axon initial segment length in setting action potential threshold and generating heterogeneity in sound-evoked responses, and thereby propose mechanisms by which GBCs may homeostatically adjust their excitability. Volume EM also reveals new dendritic structures and dendrites that lack innervation. This framework defines a pathway from subcellular morphology to synaptic connectivity, and facilitates investigation into the roles of specific cellular features in sound encoding. We also clarify the need for new experimental measurements to provide missing cellular parameters, and predict responses to sound for further in vivo studies, thereby serving as a template for investigation of other neuron classes.

## Editor's evaluation

This manuscript provides a structural analysis of bushy cells in the mouse cochlear nucleus. These neurons receive a large synaptic contact from the auditory nerve termed an endbulb that preserves the temporal information present in the auditory nerve, and are key elements of binaural sound localization circuits. The analysis combines volume electron microscopy techniques with computational models to predict heterogeneous bushy cell responses. The analysis takes morphological analysis of bushy cells to a new level, and the modeling is well done.

## Introduction

Both spherical and globular subpopulations of bushy cells (BCs) of the cochlear nucleus (CN) encode temporal fine structure and modulation of sound with high fidelity, but the globular bushy cells (GBCs) do so with greater precision. (*Bourk, 1976*; *Joris et al., 1994a*; *Joris et al., 1994b*; *Wei et al., 2017*; *Spirou et al., 1990*; *Smith et al., 1991*). GBCs, many of which are located in the auditory nerve fiber (ANF) entry zone, play central roles in hearing as they are essential for binaural processing and are a key cell type that defines and drives the early stages of the lemniscal auditory pathway (*Warr, 1966*; *Tolbert et al., 1982*; *Smith et al., 1991*; *Yin et al., 2019*; *Spirou et al., 1990*). The temporal encoding capabilities of GBCs arise from a convergence circuit motif whereby many ANFs project, via large terminals called endbulbs that contain multiple active synaptic zones, onto the cell body. (*Tolbert and Morest, 1982*; *Ryugo and Fekete, 1982*; *Ryugo and Sento, 1991*; *Ryugo et al., 1993*; *Sento and Ryugo, 1989*; *Spirou et al., 2005*; *Cant and Morest, 1979a*; *Nicol and Walmsley, 2002*; *Lauer et al., 2013*; *Held, 1893*). Furthermore, the BC membrane has low threshold K+ channels and a hyperpolarization-activated conductance (*Rothman and Manis, 2003a*; *Manis and Marx, 1991*; *Cao et al., 2007*; *Cao and Oertel, 2011*) that together constrain synaptic integration by forcing a <2ms membrane time constant and actively abbreviate synaptic potentials. This short integration time functionally converts the convergence circuit motif into either a slope-sensitive coincidence detection mechanism or a first input event detector, as tested in computational models, depending upon whether activity in the ANF terminals is subthreshold or suprathreshold (*Joris et al., 1994a*; *Rothman et al., 1993*; *Rothman and Young, 1996*). The number of convergent ANF inputs onto GBCs has been estimated using light microscopy and counted using electron microscopy (EM) for a small number of neurons (*Liberman, 1991*; *Spirou et al., 2005*). However, neither approach permits more realistic assessment of biological variance within sub- and suprathreshold populations of ANF terminals, nor their definition based on delineation of actual synaptic contacts to estimate synaptic weight. These parameters are essential for prediction of neural activity and understanding the computational modes employed by BCs.

Although the preponderance of ANF inputs are somatically targeted, the dendrites of BCs exhibit complex branching and multiple swellings that are difficult to resolve in light microscopic (LM) reconstructions (*Lorente de Nó, 1981*). Consequently, the dendritic contributions to the electrical properties of BCs have not been explored. Innervation of dendrites and soma was revealed from partial reconstruction from EM images (*Ostapoff and Morest, 1991*; *Smith and Rhode, 1987*; *Tolbert and Morest, 1982*), but values are often estimated as percent coverage rather than absolute areas. Subsampling using combined Golgi-EM histology has shown innervation of swellings and dendritic shafts (*Ostapoff and Morest, 1991*), and immunohistochemistry has further indicated the presence of at least a sparse dendritic input (*Gómez-Nieto and Rubio, 2009*). Nonetheless, a complete map of synapse location across dendrite compartments, soma, and axon has not been constructed.

To resolve these longstanding issues surrounding this key cell type, we employed volume electron microscopy (EM) in the auditory nerve entry zone of the mouse CN to provide exact data on numbers of endbulb inputs and their active zones along with surface areas of all cellular compartments. Nanoscale connectomic studies typically provide neural connectivity maps at cell to cell resolution (*Zheng et al., 2018*; *Scheffer et al., 2020*; *Turner et al., 2022*; *Bae et al., 2021*; *Shapson-Coe et al., 2021*; *Cook et al., 2019*). We extend these studies and previous modeling studies of BCs, by using detailed reconstructions from the EM images to generate and constrain compartmental models that, in turn, are used to explore mechanisms for synaptic integration and responses to temporally modulated sounds. A large range of endbulb sizes was quantified structurally, and the models predict a range of synaptic weights, some of which are suprathreshold, and responses to modeled acoustic

input that exhibit enhanced temporal processing relative to auditory nerve. The pipeline described here for compartmental model generation yields a framework to predict sound-evoked activity and its underlying cellular mechanisms, and a template on which to map new structural, molecular and functional experimental data.

## Results

### Cellular organization of the auditory nerve root region of the mouse cochlear nucleus

Despite many years of study, fundamental metrics on morphology of BC somata, dendrites and axons, and the synaptic map of innervation across these cellular compartments is far from complete. We chose volume electron microscopy (serial blockface electron microscopy (SBEM)) to systematically address these fundamental questions at high resolution and quantify structural metrics, such as membrane surface area and synaptic maps, in combination with compartmental modeling that is constrained by these measurements, to deepen our understanding of BC function. We chose the mouse for this study for three reasons. First, the intrinsic excitability, ion channel complement, and synaptic physiology of mouse bushy cells has been extensively characterized, which facilitates developing biophysically-based computational representations. Second, the mouse CN is compact, permitting the evaluation of a larger fraction of the circuit in a prescribed EM volume. Third, the tools available for mouse genetics provide an advantage for future studies to identify cells and classes of synapses, which can be mapped onto the current image volume. The image volume was taken from the auditory nerve entry zone of the mouse CN, which has a high concentration of BCs. The image volume was greater than 100 μm in each dimension and contained 26 complete BC somata and 5 complete somata of non-BCs that were likely multipolar cells (MCs; beige and rust colored, respectively, in *Figure 1*). Fascicles of ANFs coursed perpendicular to other fascicles comprised, in part, of CN axons, including those of BCs, as they exit into the trapezoid body (ANF and BC [colored mauve] axons, respectively in *Figure 1A*).

Segmentation of neurons from the image volume revealed BC somata as having eccentrically located nuclei (25/26 BCs) with non-indented nuclear envelopes (25/26 BCs; the one indented nuclear envelope was eccentrically located), and stacks of endoplasmic reticulum only along the nuclear envelope facing the bulk of the cell cytoplasm (26/26 BCs; *Figure 1B–C*). Based on these cytological criteria, location of cells in the auditory nerve root, and multiple endbulb inputs (see below), we classify these cells as globular bushy cells (GBC). We use that notation throughout the remainder of the manuscript.

Myelinated ANFs connected to large enbulb terminals synapsing onto the GBC somata. Reconstructions from volume EM permitted accurate measurement of the directly apposed surface area (ASA) between the endbulb terminal and postsynaptic membrane, and identification of synapses as clusters of vesicles along the presynaptic membrane (*Figure 1B–D and D'*). In a subset of terminals we counted the number of synapses. Because the density of synapses showed only a small decrease with increasing ASA (*Figure 1F*), we used the average density to estimate the number of synapses in each terminal and to set synaptic weights in computational models (*Figure 1F*, and see Materials and methods).

An important goal of this project was to provide accurate measurements of membrane surface areas, in order to anchor compartmental models of GBC function and facilitate comparison across species and with other cell types. We standardized a procedure based on a method to generate computational meshes (*Lee et al., 2020a*), yet preserve small somatic processes (see Materials and methods and *Figure 1—figure supplement 1*). The population of GBC somatic surface areas was slightly skewed from a Gaussian distribution (1352 (SD 168.1) μm$^2$), with one outlier (cell with indented nucleus) near 2000 μm$^2$ (*Figure 1E*). The MCs (red bars in *Figure 1E*) may represent two populations based on cells with smaller (<1700 μm$^2$) and larger (>2000 μm$^2$) somatic surface area.

### A comparison of two proposed synaptic convergence motifs for auditory nerve inputs onto globular bushy cells

With image segmentation parameters set, we next addressed competing models for synaptic organization by which GBCs can achieve higher temporal precision at the onset of sound and in phase locking to periodic stimuli than ANFs, and exhibit physiologically relevant values for spike regularity

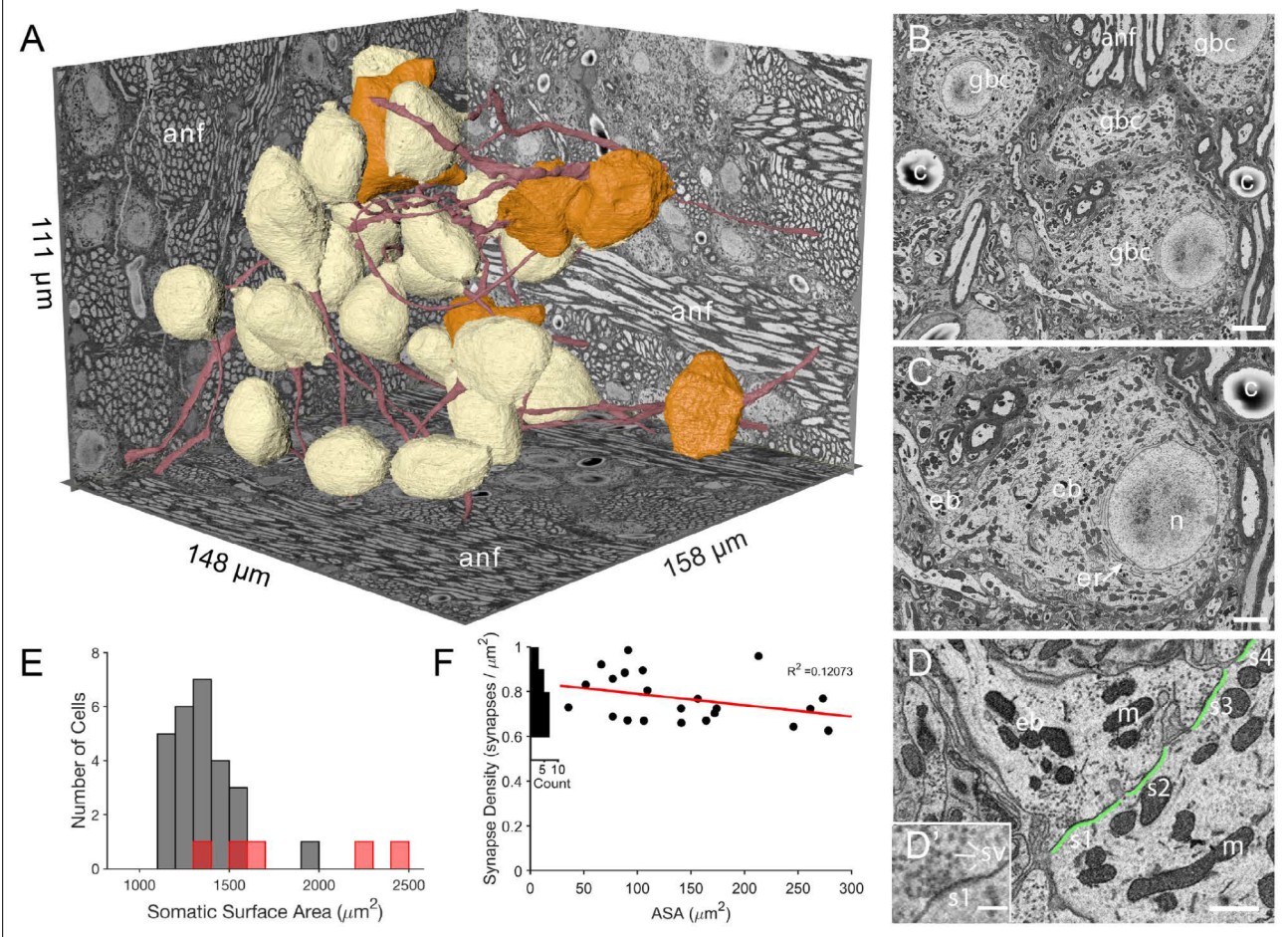

**Figure 1.** The imaged volume in the cochlear nucleus captures globular bushy (GBC) and multipolar cells (MC), and reveals synaptic sites. (**A**) The VCN region that was imaged using SBEM is depicted within walls of the image volume. Twenty-six GBCs (beige) and 5 MCs (orange) are shown with their axons (red). Left rear wall transects auditory nerve (anf) fascicles, which run parallel to the right rear wall and floor. Non-anf axons exit into the right rear wall and floor as part of other fascicles that are cross-sectioned. The complete volume can be viewed at low resolution in *Figure 1—video 1*. (**B**) Example image, cropped from the full field of view, from the data set in panel A. Field of four GBC (gbc) cell bodies, myelinated axons in anf fiber fascicles, and capillaries (c). (**C**) Closeup of the cell body (cb) of lower right GBC from panel B, illustrating the eccentrically located nucleus (**n**), short stacks of endoplasmic reticulum (er) aligned with the cytoplasm-facing side of the nuclear envelope, and contact by an endbulb (eb). (**D**) Closeup of the labeled endbulb contacting the cell in panel C (eb), revealing its initial expansion along the cell surface. Apposed pre- and postsynaptic surface area (ASA; green) are accurately determined by excluding regions with intercellular space (ASA is discontinuous), and synaptic sites (s1-4) are indicated as clusters of vesicles with some contacting the presynaptic membrane. (**D'**) Inset in panel D is closeup of synapse at lower left in panel D. It shows defining features of synapses in these SBEM images, which include clustering of vesicles near the presynaptic membrane, convex shape of the postsynaptic membrane, and in many cases a narrow band of electron-dense material just under the membrane, as evident here between the 's1' symbol and the postsynaptic membrane. Green line indicates regions of directly apposed pre- and postsynaptic membrane, and how this metric can be accurately quantified using EM. (**E**) Histogram of all somatic surface areas generated from computational meshes of the segmentation. GBCs are denoted with grey bars and MCs with red bars. (**F**) Synapse density plotted against ASA shows a weak negative correlation. Marginal histogram of density values is plotted along the ordinate. Scale bars = 5 µm in **B**, 2 µm in **C**, 1 µm in **D**, 250 nm in **D'**.

The online version of this article includes the following video and figure supplement(s) for figure 1:

**Figure supplement 1.** Steps in mesh generation and compartmental representation from EM volumes, related to *Figure 1A*.

**Figure 1—video 1.** Exploration of the relation between an image volume and a globular bushy cell (GBC) mesh derived from that volume.

https://elifesciences.org/articles/83393/figures#fig1video1

(*Rothman et al., 1993*; *Joris et al., 1994a*; *Joris et al., 1994b*). These models are based on convergence of large, somatic endbulbs of Held (*Rouiller et al., 1986*; *Liberman, 1991*; *Ryugo and Fekete, 1982*; *Figure 2A and B*). At one extreme, all convergent inputs, although harboring multiple release sites, are subthreshold for spike generation, and also of similar weight. With the functional attribute of

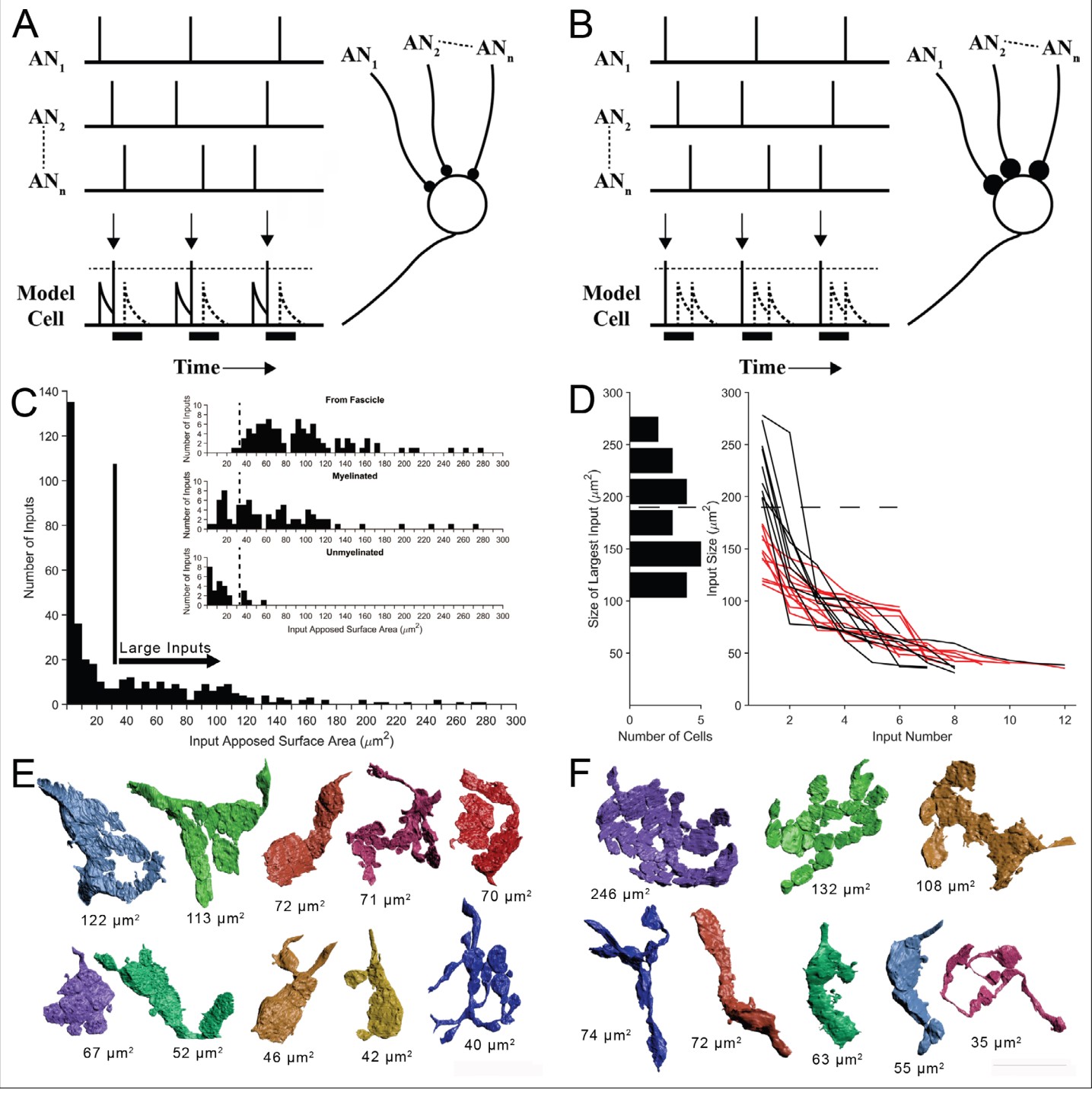

**Figure 2.** Two competing models for synaptic convergence evaluated using size profiles of endbulb terminals. (**A**) Coincidence Detection model, all inputs are subthreshold (small circles), have similar weight, and at least two inputs are active in a short temporal window to drive a postsynaptic spike. Each vertical bar is a presynaptic spike and each row is a separate auditory nerve (AN) input. Bottom line is activity of postsynaptic globular bushy cell (GBC). EPSPs are solid; action potentials indicated by vertical arrows. Dotted lines are inputs that occur during the refractory period (solid bar). Drawn after *Joris et al., 1994a*. (**B**) First-Latency model, whereby all inputs are suprathreshold (large circles), have similar weight, and the shortest latency input drives a postsynaptic spike. Longer latency inputs are suppressed during the refractory period. For a periodic sound, both models yield improved phase-locking in the postsynaptic cell relative to their auditory nerve (AN) inputs. (**C**) Histogram of input sizes, measured by apposed surface area (ASA), onto GBC somata. Minimum in histogram (vertical bar) used to define large somatic inputs (arrow). Inset: Top. Size distribution of somatic terminals traced to auditory nerve fibers within the image volume. Middle. Size distribution of somatic terminals with myelinated axons that exited the volume without being traced to parent fibers within volume. Bottom. Size distribution of somatic terminals with unmyelinated axons. Some of these axons may

*Figure 2 continued on next page*

*Figure 2 continued*

become myelinated outside of the image volume. Small terminals (left of vertical dashed lines) form a subset of all small terminals across a population of 15 GBCs. See *Figure 2—figure supplement 1* for correlations between ASA and soma areas. (**D**) Plot of ASAs for all inputs to each cell, linked by lines and ranked from largest to smallest. Size of largest input onto each cell projected as a histogram onto the ordinate. Dotted line indicates a minimum separating two populations of GBCs. Linked ASAs for GBCs above this minimum are colored black; linked ASAs for GBCs below this minimum are colored red. (**E, F**). All large inputs for two representative cells. View is from postsynaptic cell. (**E**) The largest input is below threshold defined in panel (**D**). See *Figure 2—figure supplement 2* for all 12 cells with this input pattern. (**F**) The largest input is above threshold defined in panel (**D**). All other inputs have similar range as the inputs in panel (**E**). See *Figure 2—figure supplement 3* for all 9 cells with this input pattern. Scale bar omitted because these are 3D structures with extensive curvature, and most of the terminal would be out of the plane of the scale bar. See *Figure 2—video 1* to view the somatic inputs on GBC18.

The online version of this article includes the following video and figure supplement(s) for figure 2:

**Figure supplement 1.** Morphological correlations for synapse and somatic area, related to *Figure 2C and D*.

**Figure supplement 2.** Large somatic terminals onto each globular bushy cell (GBC) that fit the Coincidence Detection model, related to *Figure 2E*.

**Figure supplement 3.** Large terminals onto each globular bushy cell (GBC) that fit the mixed Coincidence Detection/First-Arrival model, related to *Figure 2F*.

**Figure 2—video 1.** Exploration of all features of a globular bushy cell (GBC).

https://elifesciences.org/articles/83393/figures#fig2video1

a brief temporal integration window defined by the short membrane time constant, this convergence motif defines GBC operation as a coincidence detector. At the other extreme, all somatic ANF inputs are large and suprathreshold, also of similar weight. In this scenario, the GBC operates as a latency detector, such that the shortest latency input on each stimulus cycle drives the cell. In both models, the GBC refractory period suppresses delayed inputs.

In order to evaluate the predictions of these models, key metrics of the number of ANF terminal inputs and the weights of each are required. We first determined a size threshold to define endbulb terminals. All non-bouton (endbulb) and many bouton-sized somatic inputs onto 21 of 26 GBCs were reconstructed, including all somatic inputs onto two cells. We then compiled a histogram of input size based on ASA. A minimum in the distribution occurred at ~25–35 $\mu m^2$, so all inputs larger than 35 $\mu m^2$ were defined as large terminals of the endbulb class (*Figure 2C*). We next investigated whether this threshold value captured those terminals originating from ANFs, by tracing retrogradely along the axons. Terminals traced to branch locations on ANFs within the volume matched the size range of large terminals estimated from the histogram (only two were smaller than the threshold value), and were all (except one branch) connected via myelinated axons (*Figure 2C* inset, top). Nearly all axons of the remaining large terminals were also myelinated (*Figure 2C* inset, middle). The remaining few unmyelinated axons associated with large terminals immediately exit the image volume, and may become myelinated outside of the field of view (*Figure 2C* inset, bottom, right of vertical dashed line). These data together lent confidence to the value of 35 $\mu m^2$ as the size threshold for our counts of endbulb terminals. We use the terminology 'endbulb' or 'large terminal' interchangeably throughout this report.

## Five-12 auditory nerve endbulbs converge onto each globular bushy cell

After validating the size range for the endbulb class, we found a range of 5–12 convergent endings (*Figure 2D*, right). This range exceeds prior estimates of 4–6 inputs, based on physiological measures in mouse (*Cao and Oertel, 2010*). We next inquired whether the range of input size was similar across all cells. Inspecting the largest input onto each cell revealed, however, two groups of GBCs, which could be defined based on whether their largest input was greater than or less than 180 $\mu m^2$ (histogram along left ordinate in *Figure 2D*). Plotting endbulb size in rank order (largest to smallest) for each cell revealed that, excluding the largest input, the size distributions of the remaining inputs overlapped for both groups of GBCs (black and red traces in *Figure 2D*). A catalogue of all inputs for the representative cells illustrates these two innervation patterns and reveals the heterogeneity of input shapes and sizes for each cell and across the cell population (*Figure 2E, F*; *Figure 2—figure supplement 2* and *Figure 2—figure supplement 3* show all modified endbulbs for the 21 reconstructed cells). We hypothesized from this structural analysis that one group of GBCs follows the coincidence

detection (CD) model depicted in *Figure 2A* where all inputs are subthreshold (12/21 cells; red lines in *Figure 2D*), and a second group of GBCs follows a mixed coincidence-detection / latency detector model (mixed-mode, MM) where one or two inputs are suprathreshold and the remainder are subthreshold (9/21 cells; black lines in *Figure 2D*). No cells strictly matched the latency detector model (all suprathreshold inputs) depicted in *Figure 2B*.

## Innervation of globular bushy cells shows specificity for auditory nerve fiber fascicles

The majority (98/158) of end bulbs could be traced along axon branches to parent ANFs constituting fascicles within the image volume. The remaining branches exited the volume (2/6 and 3/8 branches (white arrowheads), respectively, for example cells in *Figure 3A, B*). We then asked whether the fascicle organization of the ANFs was related to innervation patterns, whereby most inputs to a particular cell might be associated with the same fascicle. We identified nine fascicles in the image volume, containing in total 1100 axons (based on a section taken through middle of volume), which is 7–15% of the total number of ANFs in mouse (7,300–16,600) (*Burda et al., 1988*; *Anniko and Arnesen, 1988*; *Camarero et al., 2001*). The largest five fascicles (containing between 115 and 260 axons/fascicle) each split into as many as seven sub-fascicles along their trajectory (*Figure 3A, B*). Excluding four cells near the edge of the image volume (GBCs 02, 24, 29, 14 plotted at left in right histogram of *Figure 3C*), 2–9 endbulbs from individual cells were traced to ANFs in the same major fascicle (for the example cells in *Figure 3A, B*, 2 fascicles each contained 2 parent axons of inputs to each cell (fascicles #2, #3, and #2, #7, respectively)). None of the parent ANFs that were linked to endbulbs branched more than once within the volume. The proportion of axons yielding endbulb terminals within the image volume was low in some fascicles (fasicles #3, #4, #5, #6; fasicle #4 contributed no endbulbs), and high in others (#1, #7; GBC08 had 9 endbulbs traced to fascicle #7). These observations indicate that the auditory nerve fascicles preferentially innervated different rostro-caudal territories of the same frequency region (*Figure 3D*).

The myelinated lengths of branches from parent fibers to terminals varied from 0 (endbulbs emerged *en passant* from parent terminal in two cases) to 133 μm (*Figure 3G*). For a subset of 10 GBCs with at least four branches traced back to parent ANFs, we utilized the resolution and advantages of volume EM to assay axon morphology. Branches were thinner than the parent ANFs, (1.4 (SD 0.33) vs 2.7 (SD 0.30) μm diameter), and both the parent ANF and branches had the same g-ratio of fiber (including myelin) to axon diameter (*Figure 3F*; ratio 0.76 across all axons). From these data, we applied a conversion of 4.6 * fiber diameter in μm (*Boyd and Kalu, 1979*; *Waxman and Bennett, 1972*) to the distribution of fiber lengths, yielding a conduction velocity range of 2.3–8.9 m/s, and a delay range of 0 (*en passant* terminal) - 15.9 μs. These values were then scaled by the $L/d$ ratio, where $L$ is the length between the ANF node and the terminal heminode, and $d$ is the axon diameter (*Brill et al., 1977*; *Waxman, 1980*). The $L/d$ ratio slows conduction velocity to a greater extent in short branches, yielding a latency range of 0-21 μs across the cell population, and a similar range among different branches to individual cells (*Figure 3G*). Such small variations in delays may affect the timing of spikes at sound onset, which can have a standard deviation of 0.39ms in mouse (*Roos and May, 2012*, measured at 30 dB re threshold, so it likely that there is a smaller SD at higher intensity), similar to values in cat (*Young et al., 1988*; *Blackburn and Sachs, 1989*; *van Gisbergen et al., 1975*; *Spirou et al., 1990*) and gerbil (*Typlt et al., 2012*). We conclude, however, that the diameter of ANF branches is sufficiently large to relax the need for accurate branch location and short-range targeting of the cell body in order to achieve temporally precise responses to amplitude-modulated or transient sounds.

## A pipeline for translating high-resolution neuron segmentations into compartmental models consistent with in vitro and in vivo data

Ten of the GBCs had their dendrites entirely or nearly entirely contained within the image volume, offering an opportunity for high-resolution compartmental modeling. The computational mesh structures of the cell surfaces (*Figure 1—figure supplement 1*), including the dendrites, cell body, axon hillock, axon initial segment, and myelinated axon were converted to a series of skeletonized nodes and radii (SWC file format *Cannon et al., 1998*; *Figure 4B*, right and *Figure 4—figure supplement 1* mesh and SWC images of all 10 cells) by tracing in 3D virtual reality software (syGlass, IstoVisio, Inc). The SWC files were in turn translated to the HOC file format for compartmental

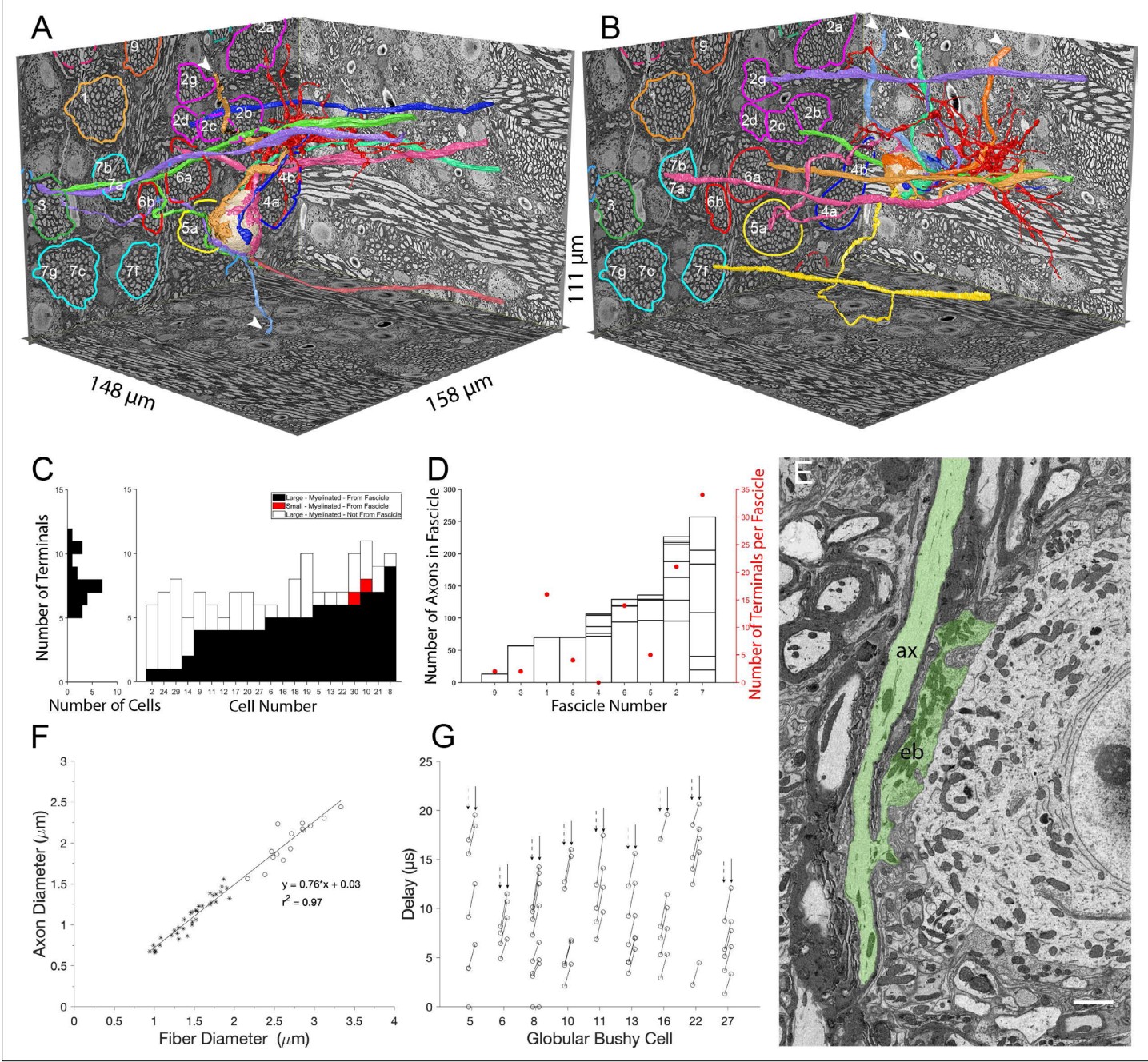

**Figure 3.** Large somatic terminals link to auditory nerve fibers (ANF) through myelinated branch axons of varying length and fascicle organization. (**A, B**) ANFs and their branches leading to all large inputs for two representative cells. ANF, branch axon and large terminal have same color; each composite structure is a different color. Convergent inputs emerge from multiple fascicles (fascicles circled and named on back left EM wall), but at least two inputs emerge from the same fascicle for each cell (green, purple axons from fascicle 3 in panel (**A**); yellow, mauve axons from fascicle 7 and green, purple from fascicle 2 in panel (**B**)). Some branch axons leave image volume before parent ANF could be identified (white arrowheads). Globular bushy cell (GBC) bodies colored beige, dendrites red, axons mauve and exit volume at back, right EM wall; axon in panel A is evident in this field of view. (**C**) Stacked histogram of branch axons traced to parent ANF (black), branch axons exiting volume without connection to parent ANF (open), small terminals linked to parent ANF (red; included to illustrate these were a minority of endings), arranged by increasing number of large terminals traced to a parent ANF per GBC. GBCs with fewest branch connections to parent ANF (GBC02, 24, 29, 14) were at edge of image volume, so most branch axons could only be traced a short distance. Number of terminals per cell indicated in horizontal histogram at left. (**D**) Number of axons in each fascicle (left ordinate) and number of axons connected to endbulb terminals per fascicle (red symbols, right ordinate). (**E**) Example of *en passant* large terminal emerging directly from node of Ranvier in parent ANF. (**F**) Constant ratio of fiber diameter (axon +myelin) / axon diameter as demonstrated by linear fit to data. All branch fiber diameters (asterisks) were thinner than ANF parent axon diameters (open circles). (**G**) Selected cells for which most branch

*Figure 3 continued on next page*

Figure 3 continued

axons were traced to a parent ANF. Lines link the associated conduction delays from parent ANF branch location for each branch, computed using the individual fiber diameters (length / conduction velocity [leftmost circle, vertical dashed arrows] or values scaled by the axon length / axon diameter [rightmost circle, vertical solid arrow]). See *Figure 3—video 1* for a detailed 3-D view of GBC11 and its inputs. Scalebar in (**E**) is 2 μm.

The online version of this article includes the following video for figure 3:

**Figure 3—video 1.** Exploration of all large somatic inputs onto a single globular bushy cell (GBC), their branch axons, and parent auditory nerve fibers.
https://elifesciences.org/articles/83393/figures#fig3video1

modeling using NEURON (*Carnevale and Hines, 2006*). The HOC versions of the cells were scaled to maintain the surface areas calculated from the meshes (see Materials and methods). An efficient computational pipeline was constructed that imported cell geometry, populated cellular compartments with ionic conductances, assigned endbulb synaptic inputs accounting for synaptic weights, and simulated the activation of ANFs for arbitrary sounds (see Materials and methods and *Figure 4C*).

Individual cell models were constructed and adjusted by mimicking in vitro measurements for $g_{KLT}$ to set channel densities (see *Figure 4—figure supplement 2*). Three models were generated for each cell, varying only in the density of channels in the dendrites. In the 'passive' model, the dendrites only had a leak conductance. In the 'active' model, the dendrites had the same channel complement and density as the soma. In the 'half-active' model, the conductances in the dendrites were set to half of the somatic density. The membrane time constant was slower by nearly a factor of 2 with the passive dendrite parameters than the active dendrite parameters, but the input resistances were very similar across the three parameter sets, with no further parameter adjustments. (*Figure 4—figure supplement 2*). All three parameter sets yielded GBC-like phasic responses to current injection, a voltage sag in response to hyperpolarizing current and a non-linear IV plot (*Figure 4D and E* and *Figure 4—figure supplement 3*). In the passive dendrite models, some cells showed trains of smaller spikes with stronger current injections, or 2–3 spikes with weaker currents (GBCs 09, 10, 11 and 30). Rebound spikes were larger and more frequent with passive dendrites than in the other two models. Rebound spikes were present in all cells with the half-active dendrite model, whereas repetitive firing was limited to 2–3 spikes, similar to what has been observed in GBCs previously (*Francis and Manis, 2000*; *Cao et al., 2007*) The active dendrite models exhibited single-spike phasic responses, and rebound action potentials were suppressed (GBCs 05, 06, and 10) or smaller in amplitude. Because the differences in intrinsic excitability were modest across the models, and because the half-active dendrite model most closely resembled typical responses reported in vitro, we used the half-active dendrite models for the remainder of the simulations.

Next, we investigated the responses to simulated sound inputs. For these simulations, the number of synapses in each endbulb was based on the endbulb ASA and the average synapse density (*Figure 1F*). Terminal release was simulated with a stochastic multi-site release model in which each synapse in the terminal operated independently (*Xie and Manis, 2013b*; *Manis and Campagnola, 2018*). Synaptic conductances were not tuned, but instead calculated based on experimental measurements as described previously (*Manis and Campagnola, 2018*). Action potentials (AP; marked by red dots in *Figure 4D and F*) were detected based on amplitude, slope and width at half-height (*Hight and Kalluri, 2016*). ANFs were driven in response to arbitrary sounds via spike trains derived from a cochlear model (*Zilany et al., 2014*; *Rudnicki et al., 2015*; *Figure 4C*, right). As expected, these spike trains generated primary-like (Pri) responses in ANFs and yielded Pri or primary-like with notch (Pri-N) responses in the GBC models (*Figure 4F–G*; *Figure 4—figure supplement 4*). The predicted SD of the first spike latency in the model varied from 0.232 to 0.404ms (*Figure 4—figure supplement 4*), while the coefficient of variation of interspike intervals ranged from 0.45 to 0.73. These ranges are similar to values reported for mouse CN in vivo (*Roos and May, 2012*). Taken together, these simulations, which were based primarily on previous electrophysiological measurements and the volume EM reconstructions, without further adjustments, produced responses that are quantitatively well-matched with the limited published data. Using these models, we next explored the predicted contributions of different sized inputs and morphological features to spike generation and temporal coding in GBCs.

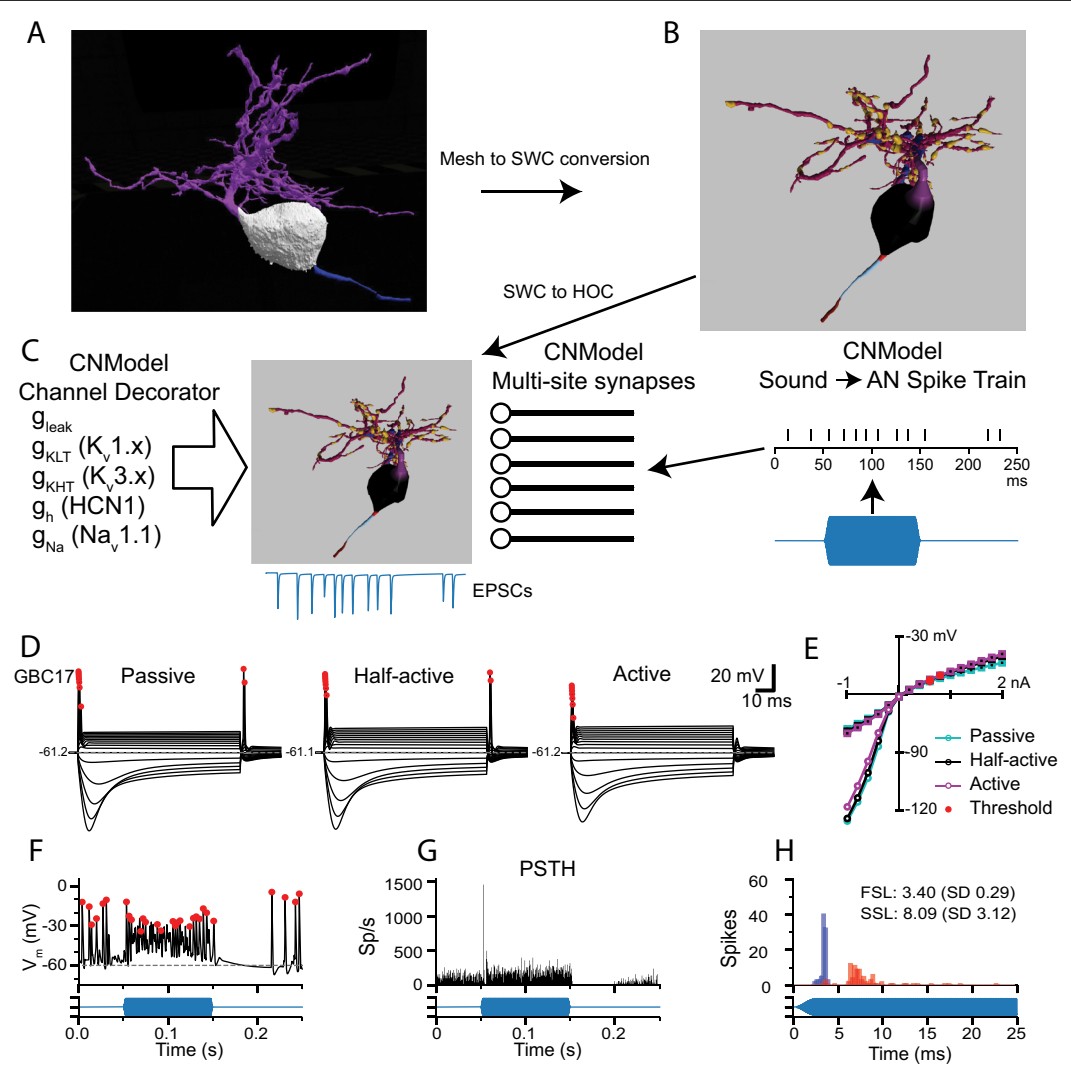

**Figure 4.** Pipeline to generate compartmental models for analysis of synaptic integration and electrical excitability from the mesh reconstructions of mouse VCN bushy neurons. (**A**) The mesh representation of the volume EM segmentation was traced using syGlass virtual reality software to generate an SWC file consisting of locations, radii, and the identity of cell parts (**B**). In (**B**), the myelinated axon is dark red, the axon initial segment is light blue, the axon hillock is red, the soma is black, the primary dendrite is purple, dendritic hubs are blue, the secondary dendrite is dark magenta, and the swellings are gold. The mesh reconstruction and SWC reconstructions are shown from different viewpoints. See *Figure 1—figure supplement 1* for all reconstructions. See *Figure 4—video 1* for a 3D view of the mesh and reconstructions for GBC11. (**C**) The resulting SWC model is decorated with ion channels (see *Figure 4—figure supplement 2* for approach), and receives inputs from multi-site synapses scaled by the apposed surface area of each ending. For simulations of auditory nerve input, sounds (blue) are converted to spike trains to drive synaptic release. (**D**). Comparison of responses to current pulses ranging from –1 to +2 nA for each dendrite decoration scheme. In the Passive scheme, the dendrites contain only leak channels; in the Active scheme, the dendrites are uniformly decorated with the same density of channels as in the soma. In the Half-active scheme, the dendritic channel density is one-half that of the soma. (**E**) Current voltage relationships for the 3 different decoration schemes shown in (**D**). Curves indicated with circles correspond to the peak voltage (exclusive of APs); curves indicated with squares correspond to the steady state voltage during the last 10ms of the current step. Red circles indicate the AP threshold. (**F**) Example of voltage response to a tone pip in this cell (Half-active model). Action potentials are marked with red dots, and are defined by rate of depolarization and amplitude (see Methods). (**G**) Peri-stimulus time histogram (PSTH) for 50 trials of responses to a 4 kHz 100 ms duration tone burst at 30 dB SPL. The model shows a primary-like with notch response. See *Figure 4—figure supplement 4* for all tone burst responses. (**H**) First spike latency (FSL; blue) and second spike latency (SSL; red) histograms for the responses to the tone pips in G. (**F,G,H**) The stimulus timing is indicated in blue, below the traces and histograms.

*Figure 4 continued on next page*

*Figure 4 continued*

The online version of this article includes the following video and figure supplement(s) for figure 4:

**Figure supplement 1.** Segmented globular bushy cells (GBC) and their representations for compartmental modeling, related to *Figure 4A and B*.

**Figure supplement 2.** Conductance scaling using voltage clamp simulations for different patterns of dendrite decoration, related to *Figure 4C*.

**Figure supplement 3.** Current-clamp responses for all 10 complete bushy cells for each of the ion channel decoration conditions, related to *Figure 4D and E*.

**Figure supplement 4.** Peri-stimulus time histograms (PSTH), spike latencies and interpsike interval regularity in response to tone bursts at characteristic frequency, related to *Figure 4F–H*.

**Figure 4—video 1.** Comparison of cell structures and regions between a 3D mesh and an SWC representation of a globular bushy cell (GBC).

https://elifesciences.org/articles/83393/figures#fig4video1

## Model predictions

The individual GBCs showed variation in the patterns of endbulb size, dendrite area and axon initial segment length. In this section, we examine the model predictions for each of the fully reconstructed GBCs to address five groups of predictions about synaptic integration and temporal precision in GBCs.

### Prediction 1: Endbulb size does not strictly predict synaptic efficacy

The wide variation in size of the endbulb inputs (*Figure 2C–F*) suggests that inputs with a range of synaptic strengths converge onto the GBCs. We then inquired whether individual cells followed the coincidence-detection or mixed-mode models hypothesized by input sizes shown in *Figure 2D*. To address this question, we first modeled the responses by each of the 10 fully reconstructed GBCs as their endbulb inputs were individually activated by spontaneous activity or 30 dB SPL, 16 kHz tones (responses at 30 dB SPL for four representative cells (GBC05, 30, 09, and 17 are shown in *Figure 5A*; the remaining GBCs are shown in *Figure 5—figure supplement 1*)). In *Figure 5A*, voltage responses to individual inputs are rank-ordered from largest (1) to smallest (7,8,or 9) for each cell. Without specific knowledge of the spontaneous rate or a justifiable morphological proxy measure, we modeled all ANFs as having high spontaneous rates since this group delivers the most contacts to GBCs in cat (Figure 9 in *Liberman, 1991*).

We chose four cells to illustrate the range of model responses. GBC05 and GBC30 (*Figure 5A1 and A2*) fit the coincidence-detection model, in that none of their inputs individually drove post-synaptic APs except the largest input for GBC30, which did so with very low efficacy (#postsynaptic APs/#presynaptic APs; see also GBC10, GBC06, GBC02, GBC13 in *Figure 4—figure supplement 1*). GBC09 and GBC17 (*Figure 5A3 and A4*) fit the mixed-model, in that the largest inputs (2 large inputs for GBC17) individually drive APs with high efficacy (see also GBC11, GBC18 in *Figure 4—figure supplement 1*). This result demonstrates two populations of GBCs based on the absence or presence of high efficacy suprathreshold inputs.

The second largest input for GBC09 (132 $\mu m^2$) had higher efficacy than the largest input for GBC30 (172 $\mu m^2$). The variation of efficacy for similar ASA was evident, especially between 125–175 $\mu m^2$, in a plot of all inputs across the ten GBCs (*Figure 5D*). Since many cells lacked inputs in this range, we created 3 different sizes of artificial synapses (150, 190 and 230 $\mu m^2$) onto GBCs 10, 17 and 30 to predict the efficacy of a more complete range of input sizes. The addition of these inputs (stars colored for each cell) reinforced the suggestion that there were two populations of GBCs, of greater (GBCs 09, 11, 13, 17; red curve) or lesser excitability (GBCs 02, 05, 06, 10, 18, 30; cyan curve). Therefore, we combined all synapses (excluding the artificial synapses) from GBCs 09, 11, 13, and 17 into one group, and synapses from all the remaining cells, GBCs 02, 05, 06, 10, 18 and 30, into a second group. GBC18 was included in the lesser excitability group event though it had a single large input, because all of its smaller inputs grouped with the input efficacy for the other cells with lower excitability. We then confirmed the efficacy data by fitting each group with logistic functions with distinct parameters (*Figure 5D*). The group with the greater excitability had half-maximal size for input ASAs of 148.6 (SD 1.1) $\mu m^2$ and a maximal efficacy of 0.72 (SB 0.01) $\mu m^2$, with a slope factor of 14.3 (SD 1.1)/

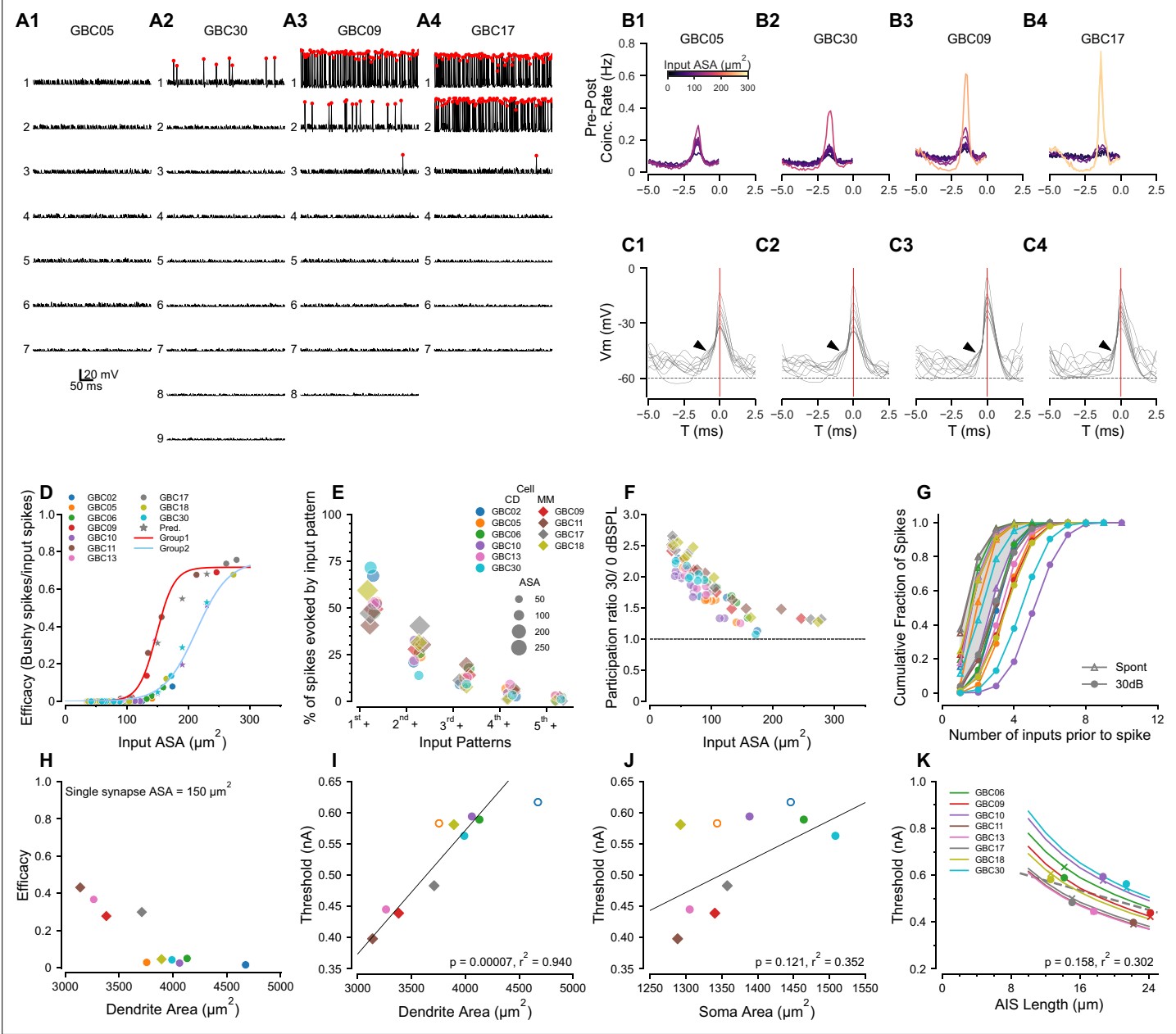

**Figure 5.** Compartmental models predict sub- and suprathreshold inputs, efficacy dependence on dendrite surface area, and rate-dependent participation in spike generation. (A1-4) Simulations showing EPSPs and spikes in response to individual ANFs in 4 model globular busjy cells (GBCs) during a 30 dB SPL tone pip, arranged by efficacy of the largest input. Spikes indicated by red dots. Vertically, traces are ranked ordered by endbulb size. Responses for the other six cells are shown in *Figure 5—figure supplement 1*. (B1-4) Cross-correlations between postsynaptic spikes and spikes from each input ANF during responses to 30 dB SPL tones (all inputs active). Trace colors correspond to the ASA of each input (color bar in (**B1**)). See *Figure 5—figure supplement 1* for cross-correlations for the other 6 cells. (C1-4): Voltage traces aligned on the spike peaks for each of the 4 cells in (**B**). Postsynaptic spikes without another spike within the preceding 5ms were selected to show the subthreshold voltage trajectory more clearly. Zero time (0ms; indicated by vertical red line) is aligned at the action potential (AP) peak and corresponds to the 0 time in (B1-4). Arrowheads indicate EPSPs preceding the SP in panels (C1-3); arrowhead in C4 shows APs emerging directly from the baseline, indicating suprathreshold inputs. (**D**) GBCs could be divided into two groups based on the pattern of efficacy growth with input size. GBCs 09, 11, 13 and 17 formed one group, and GBCs 02, 05, 06, 10, 18, and 30 formed a second group with overall lower efficacy. The red line is a best fit logistic function to the higher efficacy group. The blue line is the logistic fit to the lower efficacy group. Stars indicate test ASA-efficacy points, supporting membership in the lower efficacy group for cells 10 and 30. (**E**) Comparison of the patterns of individual inputs that generate spikes. Ordinate: $1^{st}+$ indicates spikes driven by the largest input plus any other inputs. $2^{nd}+$ indicates spikes driven by the second largest input plus any smaller inputs, excluding spikes in which the largest input was active. $3^{rd}+$ indicates spikes driven by the third largest input plus any smaller inputs, but not the first and second largest inputs. $4^{th}+$ indicates contributions from

*Figure 5 continued on next page*

*Figure 5 continued*

the fourth largest input plus any smaller inputs, but not the 3 largest. $5^{th}+$ indicates contributions from the fifth largest input plus any smaller inputs, but not the 4 largest. Colors and symbols are coded to individual cells, here grouped according to predicted coincidence mode or mixed-mode input patterns as shown in *Figure 2—figure supplement 2* and *Figure 2—figure supplement 3*. See *Figure 5—figure supplement 2* for a additional summaries of spikes driven by different input patterns. (**F**) The participation of weaker inputs (smaller terminal area) is increased during driven activity at 30 dB SPL relative to participation during spontaneous activity. The dashed line indicates equal participation at the two levels. Each input is plotted separately. Colors and symbols are coded to individual cells as in (**D**). (**G**) Cumulative distribution of the number of inputs driving postsynaptic spikes during spontaneous activity and at 30 dB SPL. The color for each cell is the same as in the legend in (**D**). Symbols correspond to the stimulus condition. (**H**) Efficacy for a single 150 µm² input is inversely related to dendrite surface area. (**I**) Dendrite area and action potential threshold are highly correlated. Open circles (GBC02 and GBC05) indicate thresholds calculated using the average AIS length, but are not included in the regression. Colors and symbols as in (**D**). (**J**) Soma area and action potential threshold are not well correlated. Open symbols are as in (**I**). (**K**) Variation of AIS length using the averaged axon morphology reveals an inverse relationship to spike threshold for all cells (lines). Crosses indicate thresholds interpolated onto the lines for the averaged axon simulations; circles indicate thresholds measured in each cell with their own axon. Cells GBC02 and GBC05 are omitted because AIS length is not known. Across the cell population, the thresholds are only weakly correlated with AIS length (linear regression indicated by dashed grey line). Colors as in (**E**).

The online version of this article includes the following figure supplement(s) for figure 5:

**Figure supplement 1.** Cross-correlation plots for six additional modeled cells, related to *Figure 5A–C*.

**Figure supplement 2.** Contributions of different input patterns to postsynaptic spiking, related to *Figure 5E*.

---

µm². The fit to the group with lesser excitability (*Figure 5D*, light blue line) yielded a half-maximal size of 204.3 (SD 4.7) µm², and with a slope factor of 19.8 (SD 2.2)/µm². Cells with lesser and greater excitability were found in both the coincidence-detection (lesser: GBC02, 05, 06, 10 30; greater: GBC13) and mixed-mode (lesser: GBC18; greater GBC09, 11, 17) categories described above. Additional factors that affect excitability are discussed below in connection with Predictions 3 and 4.

## Prediction 2: Mixed-mode cells operate in both latency and coincidence-detection modes when all inputs are active

The predicted grouping of cells according to synaptic efficacy of individual inputs raises the question of how these cells respond when all inputs are active. In particular, given the range of synapse sizes and weights, we considered the contribution of the smaller versus larger inputs even within coincidence detection size profiles. To address this question, we computed GBC responses when all ANFs to a model cell were driven at 30 dB SPL and active at the same average rate of 200 Hz. We then calculated the cross-correlation between the postsynaptic spikes and each individual input occurring within a narrow time window before each spike. These simulations and cross-correlations are summarized in *Figure 5B–C*, for the four cells shown in *Figure 5A*, and in *Figure 5—figure supplement 1* for the other six cells.

For GBC05 and GBC30, which had no suprathreshold inputs, all inputs had low coincidence rates. However, not all inputs had equal contribution in that the largest input had a rate 3–4 times the rate of the smallest input (*Figure 5B1 and B2*). In both cells, the requirement to integrate multiple inputs was evident in voltage traces exhibiting EPSPs preceding an AP (*Figure 5C1 and C2*). GBC09 and GBC17 illustrate responses when cells have one or two secure suprathreshold inputs, respectively (*Figure 5A3 and A4*). The cross-correlation plots reveal the dominance of high probability suprathreshold inputs in generating APs in GBCs (yellow traces for GBC09, 17). For GBC09 but not GBC17 (likely because GBC17 has two suprathreshold inputs), all subthreshold inputs had appreciable coincidence rates. The summation of inputs to generate many of the APs for GBC09 is seen in the voltage traces preceding spikes, but most APs for GBC17 emerge rapidly without a clear preceding EPSP (*Figure 5C3 and C4*, respectively).

To understand how weaker inputs contributed independently of the largest inputs, we also calculated the fraction of postsynaptic spikes that were generated without the participation of simultaneous spikes from the N larger inputs (where N varied from 1 to number of inputs - 1, thus successively peeling away spikes generated by the larger inputs). We focused initially on mixed-mode cells (*Figure 5E*). We first calculated the fraction of postsynaptic spikes generated by the largest input in any combination with other inputs (in the time window –2.7 to –0.5ms relative to the spike peak as in *Figure 5B*). This fraction ranged from 40% to 60% in mixed-mode cells (hexagons, $1^{st}+$ in *Figure 4E*). The fraction of postsynaptic spikes generated by the second-largest input in any combination with

other smaller inputs was surprisingly large, ranging from 25% to 30% (excluding GBC17 which had 2 suprathreshold inputs; $2^{nd}$+ in *Figure 5E*). Notably, all combinations of inputs including the 3rd largest and other smaller inputs accounted for about 25% of all postsynaptic spikes. Thus, a significant fraction (about 50%) of postsynaptic spikes in mixed-mode cells are predicted to be generated by various combinations of subthreshold inputs operating in coincidence detection mode.

For GBCs that are predicted to operate in the coincidence-detection mode, we hypothesized that the contributions of different sized inputs would be more uniform. We tested this using tone stimuli at 30 dB SPL. Surprisingly, in two of the cells with the largest inputs (GBC02, GBC30), the largest input in combination with all of the smaller inputs (circles, $1^{st}$+ in *Figure 5E*) accounted for a larger percentage of postsynaptic spikes than in any of the mixed-mode cells. Notably, the largest inputs for these two cells could individually drive postsynaptic spikes, but at very low efficacy. Across the remaining cells, the $1^{st}$+ category accounted for about 50% of all postsynaptic spikes similar to the mixed model cells. These simulations thus predict that, even among coincidence detection profiles, the contributions by individual endbulbs to activity vary greatly, whereby larger inputs can have a disproportional influence that equals or exceeds that of suprathreshold inputs in mixed-mode cells.

We next inquired whether the participation of weak inputs in AP generation depended on stimulus intensity (spontaneous activity at 0 dB SPL and driven activity at 30 dB SPL), or was normalized by the increase in postsynaptic firing rate. To address this question, we computed a participation metric for each endbulb as #postsynaptic APs for which a presynaptic AP from a given input occurred in the integration window (−2.7 to −0.5ms relative to the spike peak), divided by the total number of #postsynaptic APs. The smaller inputs have a higher relative participation at 30 dB SPL than larger inputs (*Figure 5F*), suggesting a rate-based increase in coincidence among weaker inputs at higher intensities. This level-dependent role of smaller inputs was also explored in cumulative probability plots of the number of inputs active prior to a spike between spontaneous and sound-driven ANFs. During spontaneous activity, often only one or two inputs were active prior to an AP (*Figure 4G*, triangles). However, during tone-driven activity postsynaptic spikes were, on average, preceded by coincidence of more inputs (*Figure 5G*, filled circles). This leads to the prediction that mixed-mode cells depend on the average afferent firing rates of the individual inputs (sound level dependent), and the specific distribution of input strengths. Furthermore, GBCs operating in the coincidence-detection mode show a similar participation bias toward their largest inputs.

## Prediction 3: Dendrite surface area is an important determinant of globular bushy cell excitability

Although the synaptic ASA distribution plays a critical role in how spikes are generated, the response to synaptic input also depends on postsynaptic electronic structure, which determines the patterns of synaptic and ion channel-initiated current flow across the entire membrane of the cell. To further clarify how differences in excitability depend on the cell morphology, we examined the relationship between somatic and dendritic surface areas, and cellular excitability. The GBC dendrite surface area spanned a broad range from 3000–4500 $\mu m^2$. Interestingly, the GBCs having the smallest dendrite surface area comprised the group with the greatest excitability as measured by current threshold and the efficacy of a standardized 150$\mu m^2$ input (*Figure 5H*), predicting an important mechanism by which GBCs can modulate their excitability. The large difference in excitability between GBC17 and GBC05 (*Figure 5H*), which have similar surface areas, indicates that other mechanisms, perhaps related to dendritic branch patterns, are needed to explain these data fully.

To explore contributions of cell geometry to synaptic efficacy, we plotted threshold as a function of compartment surface area or length. Threshold was highly correlated with dendrite surface area ($p < 0.001$, $r^2 = 0.94$, *Figure 5I*), but modestly correlated with soma surface area ($p < 0.121$, $r^2 = 0.352$, *Figure 5J*) or the ratio of dendrite to soma surface areas ($p < 0.046$, $r^2 = 0.511$). Taken together, these simulations predict that dendrite surface area is a stronger determinant of excitability than soma surface area and that excitability is not correlated with innervation category (coincidence detection or mixed mode), under the assumption that ion channel densities are constant across cells.

## Prediction 4: Axon initial segment length modulates globular bushy cell excitability

Another factor that can regulate excitability is the length of the AIS. Therefore, in the EM volume we also quantified the lengths of the axon hillock, defined as the taper of the cell body into the axon, and the axon initial segment (AIS), defined as the axon segment between the hillock and first myelin heminode. The axon hillock was short (2.3 (SD 0.9) µm measured in all 21 GBCs with reconstructed endbulbs). The AIS length averaged 16.8 (SD 6.3) µm (range 14.2-21.4 µm ; n=16, the remaining five axons exited the volume before becoming myelinated) and was thinner than the myelinated axon. Because the conductance density of $Na^+$ channels was modeled as constant across cells, the AIS length potentially emerges as a parameter affecting excitability. To characterize this relationship, in the 10 GBCs used for compartmental modeling, we replaced the individual axons with the population averaged axon hillock and initial myelinated axon, and systematically varied AIS length. Indeed, for each cell the threshold to a somatic current pulse decreased by nearly 40% with increasing AIS length across the measured range of values (*Figure 5K*). Although threshold varied by cell, the current threshold and AIS length were not significantly correlated (p < 0.158, $r^2$ = 0.302, *Figure 5K*). These simulations predict that AIS length and dendrite area together serve as mechanisms to tune excitability across the GBC population, although dendrite area appears to have a greater contribution.

In 20 of 21 cells for which all large inputs were reconstructed, at least one endbulb terminal (range 1–4) extended onto the axon (hillock and/or the AIS), contacting an average of 18.5 (SD 10) $µm^2$ of the axonal surface (range 0.7-35.2 $µm^2$ ). The combined hillock/initial segment of every cell was also innervated by 11.8 (SD 5.6) smaller terminals (range 4-22; n=16). These innervation features will be further explored once the excitatory and inhibitory nature of the inputs, and the SR of endbulb terminals are better understood.

## Prediction 5: Temporal precision of globular bushy cells varies by distribution of endbulb size

Auditory neurons can exhibit precisely-timed spikes in response to different features of sounds. Mice can encode temporal fine structure for pure tones at frequencies only as low as 1 kHz, although with vector strength (VS) values comparable to larger rodents such as guinea pigs (*Taberner and Liberman, 2005*; *Palmer and Russell, 1986*). However, they do have both behavioral (*Cai and Dent, 2020*) and physiological (*Kopp-Scheinpflug et al., 2003*; *Walton et al., 2002*) sensitivity to sinusoidal amplitude modulation (SAM) in the range from 10 to 1000 Hz on higher frequency carriers. As amplitude modulation is an important temporal auditory cue in both communication and environmental sounds, we used SAM to assess the temporal precision of GBC spiking, which has been reported to exceed that of ANFs (*Joris et al., 1994a*; *Louage et al., 2005*; *Frisina et al., 1990*). Because temporal precision also exists for transient stimuli, we additionally used click trains. Given the variation of mixed-mode and coincidence-detection convergence motifs across GBCs, we hypothesized that their temporal precision would differ across frequency and in relation to ANFs. The left columns of *Figure 6* illustrate the flexibility of our modeling pipeline to generate and analyze responses to arbitrary complex sounds in order to test this hypothesis. SAM tones were presented with varying modulation frequency and a carrier frequency of 16 kHz at 15 and 30 dB SPL (see *Figure 6—figure supplement 1* for comparison of SAM responses in ANFs and a simple GBC model used to select these intensities), and 60 Hz click trains were presented at 30 dB SPL. We implemented a standard measure of temporal fidelity (vector strength) for SAM stimuli. To analyze temporal precision of click trains, we used the less commonly employed shuffled autocorrelogram (SAC) metric, which removes potential contribution of the AP refractory period to temporal measures (*Louage et al., 2004*).

Here, we illustrate a representative range of cellular responses and analytics available in our pipeline, from intracellular voltage traces (*Figure 6A and H*) recorded in any cellular compartment (cell body depicted here), to event data with associated representations as raster plots and period histograms. GBCs exhibited a more temporally constrained distribution of spikes in response to SAM tones and click trains (*Figure 6B–F1–M*, respectively, shown for GBC17) relative to ANFs. Measures of temporal precision demonstrate an improvement between ANFs and GBC responses to SAM tones (higher VS in *Figure 6F*). The responses to clicks consist of well-timed spikes, followed by a short refractory period before the ANF spontaneous activity recovers and drives the cell (*Figure 6J and L*). The precision of responses to clicks is also better (narrower SAC half-width) in the GBCs than in their

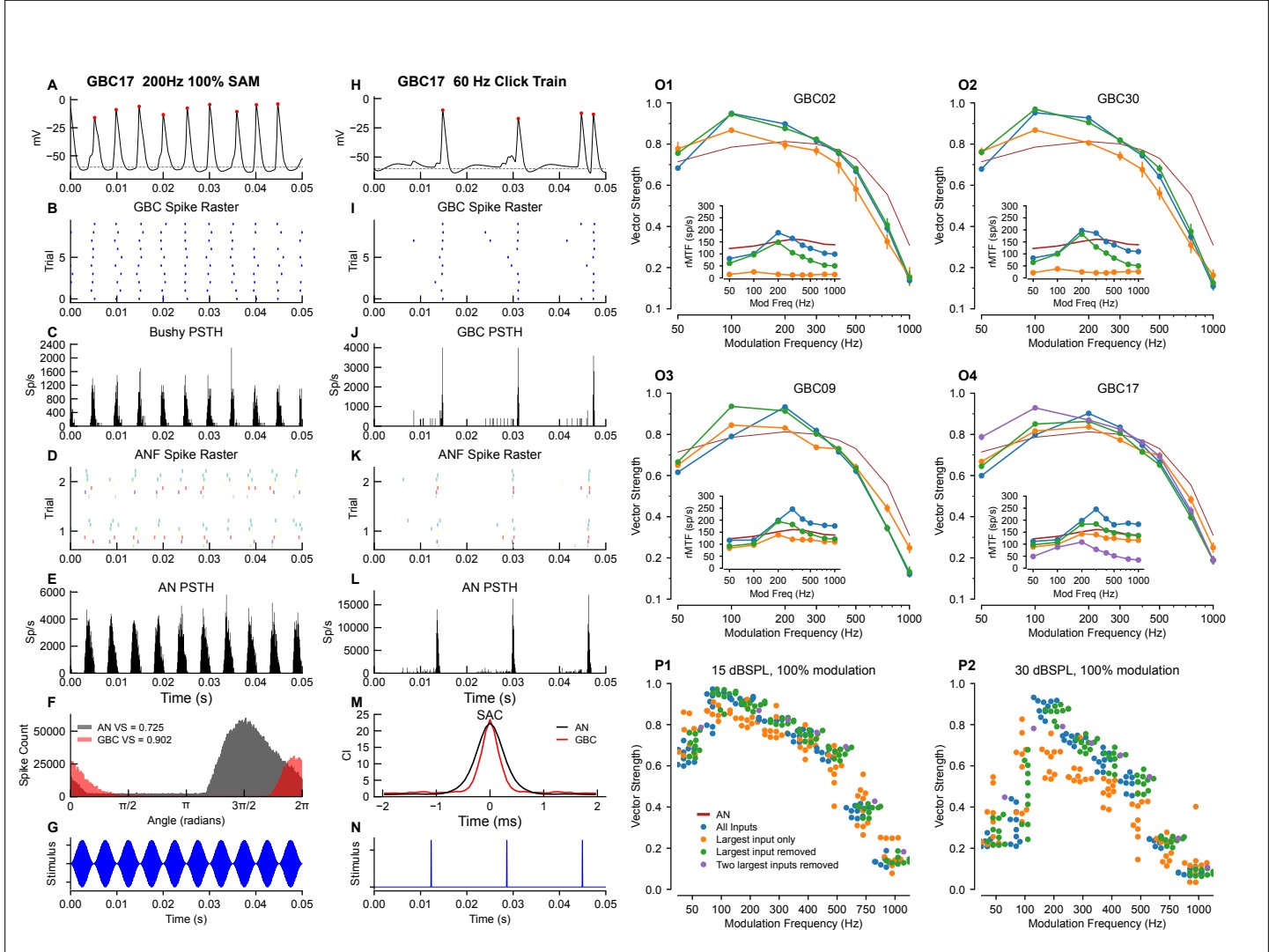

**Figure 6.** Temporal and rate modulation transfer functions and entrainment to clicks can exceed ANF values and differ between coincidence detection and mixed mode cells. Left column (**A–G**): Example of entrainment to 100% modulated SAM at 200 Hz, at 15 dB SPL. The sound level was chosen to be near the maximum for phase locking to the envelope in ANFs (see *Figure 6—figure supplement 1*). (**A**) Voltage showing spiking during a 150ms window starting 300ms into a 1 second long stimulus. (**B**) Spike raster for 100 trials shows precise firing. (**C**) PSTH for the spike raster in (**B**). (**D**) Spike raster for all ANF inputs across a subset of 2 trials. Inputs are color coded by ASA. (**E**) PSTH for the ANF. (**F**) Superimposition of the phase histograms for the GBC (black) and all of its ANF inputs (red). (**G**) Stimulus waveform. Center column (**H–N**): responses to a 50 Hz click train at 30 dB SPL. (**H**) GBC membrane potential. (**I**) Raster plot of spikes across 25 trials. (**J**) PSTH showing spike times from I. (**K**) ANF spike raster shows the ANFs responding to the clicks. (**L**) PSTH of ANF firing. (**M**) The shuffled autocorrelation index shows that temporal precision is greater (smaller half-width) in the GBC than in the ANs. See *Figure 6—figure supplement 4* for SAC analysis of other cells. (**N**) Click stimulus waveform. Right column (**O–P**): Summary plots of vector strength. (O1-4) Vector strength as a function of modulation frequency at 15 dB SPL for 3 (4 for GBC17) different input configurations. Vertical lines indicate the SD of the VS computed as described in the Methods. Insets show the rate modulation transfer function (rMTF) for each of the input configurations. Red line: average ANF VS and rMTF (insets). See *Figure 6—figure supplement 2* for the other cells. *Figure 6—figure supplement 3* shows spike entrainment, another measure of temporal processing. The legend in (**P1**) applies to all panels in (**O**) and (**P**). (**P**) Scatter plot across all cells showing VS as a function of modulation frequency for 3 (4 for GBC17) different input configurations. (**P1**) VS at 15 dB SPL. (**P2**) VS at 30 dB SPL.

The online version of this article includes the following figure supplement(s) for figure 6:

**Figure supplement 1.** Spike synchronization to stimulus envelope as a function of average stimulus intensity in ANF inputs, related to *Figure 6F and P1 and P2*.

**Figure supplement 2.** Vector strength of the 6 other globular bushy cells (GBCs) in response to 100% SAM tones at frequencies from 50 to 1000 Hz on a 16kHz carrier at 15 dB SPL, related to *Figure 6O1-4*.

**Figure supplement 3.** Spike entrainment across all globular bushy cells (GBCs) at 15 and 30 dB SPL when different combinations of inputs are active, related to *Figure 6P1-2*.

*Figure 6 continued on next page*

*Figure 6 continued*

**Figure supplement 4.** Shuffled autocorrelations (SACs) in response to click trains show importance of weaker inputs in improving temporal precision, related to *Figure 6M*.

ANF inputs (*Figure 6M*). We then compared responses of GBCs to ANFs across a range of modulation frequencies from 50 to 1000 Hz at 15 dB SPL, which revealed the tuning of GBCs to SAM tones. GBCs had higher VS at low modulation frequencies (<300 Hz), and lower VS at higher modulation frequencies (>300 Hz). Responses varied by convergence motif, whereby coincidence-detection GBCs had enhanced VS relative to ANFs at 100 and 200 Hz (*Figure 6O1–O2*, GBC02 and GBC30), but mixed-mode GBCs only at 200 Hz (blue lines in *Figure 6 O3–O4*, GBC09 and GBC17).

We explored the tuning of GBCs innervated by mixed mode and coincidence detection input profiles to the modulation frequency of SAM tones by manipulating the activation of endbulbs for each cell. At a modulation frequency of 100 Hz, inputs were dispersed in time so that combinations of small inputs and suprathreshold inputs could generate spikes at different phases of modulation. We hypothesized that removing the largest input and, for GBC17, the two largest inputs, would convert mixed mode into coincidence detection profiles. Indeed, this modification improved VS at 50 and 100 Hz, and the tuning profile broadened to resemble the coincidence detection GBCs (green, purple traces in *Figure 6 O3-4*). The same manipulation of removing the largest input for coincidence detection cells did not change their tuning, except for a small increase in VS at the lowest modulation frequency (50 Hz). Conversely, we hypothesized that removing all inputs except the largest input for mixed mode cells would make the GBCs more similar to ANFs, because they could follow only the single suprathreshold input. In this single input configuration, VS decreased at low modulation frequency and increased at high modulation frequency, making the tuning more similar to ANFs (orange traces in *Figure 6 O3-4*). A similar manipulation for coincidence detection input profiles, in which the largest input was able to drive postsynaptic spikes only with low probability (largest inputs of the other coincidence detection neurons did not drive spikes in their GBC), decreased the VS at 100 and 200 Hz, but also decreased VS for modulation frequencies ≥300 Hz. The consistency across cells of changes in modulation sensitivity with these manipulations can also be appreciated across all cells as plotted in *Figure 6P1 and P2*.

We also computed the rate modulation transfer functions (rMTF) for each input configuration (insets in *Figure 6 O1-4* and *Figure 6—figure supplement 2A-F*). For coincidence-detection neurons these functions have a band-pass shape, peaking at 200–300 Hz for configurations with all inputs and configurations lacking the largest input. On the other hand, the largest input alone results in low firing rates. For mixed-mode cells, the rMTF is more strongly bandpass and has a higher rate with all inputs, or all inputs except the largest, whereas the rates are lower and the bandpass characteristic is less pronounced with the largest input alone.

Entrainment, the ability of a cell to spike on each stimulus cycle (see Materials and methods for calculation), was predicted to be better than entrainment in the ANFs up to about 300 Hz (*Figure 6—figure supplement 3A and B*) for all GBCs with all inputs for the coincidence-detection neurons. Entrainment dropped to low values at 500 Hz and above. Entrainment for mixed-mode cells exceeded that of coincidence-detection cells, and was nearly equal to that of ANFs up to 200 Hz (*Figure 6—figure supplement 3C and D*). Entrainment for all cells except GBC02 exceeded that of the ANF up to 200 Hz in the absence of the largest input (*Figure 6—figure supplement 3E and F*).

Similarly, improvements in temporal precision were evident in response to click trains *Figure 6—figure supplement 4*. The half-widths of the SACs (when there were sufficient spikes for the computation) were consistently narrower and had higher correlation indices when all inputs, or all but the largest input were active, than when only the largest input was active. The coincidence-detection GBCs showed the highest correlation indices and slightly narrower half-width (*Figure 6—figure supplement 4*). Taken together, the different convergence motifs yielded a range of tuning (mixed-mode GBCs more tuned) to the modulation frequency of SAM tones in comparison to ANFs. Notably, the mixed-mode GBCs with the most pronounced tuning were those whose inputs most easily excited their postsynaptic GBC (*Figure 5*), because their response at 100 Hz was similar to that of ANFs. Thus, the ANF convergence patterns play an important role in setting the temporal precision of individual GBCs.

## Globular bushy cell dendrites exhibit non-canonical branching patterns and high-degree branching nodes

GBC dendrites have been noted to have dense branching such that they elude accurate reconstruction using light microscopy (*Lorente de Nó, 1981*). Volume EM permitted full and accurate reconstructions, which revealed novel features. Of the 26 GBCs, 24 extended a single proximal dendrite (although one dendrite branched after 1.8 m), and 2 extended two proximal dendrites. Proximal dendrite length was measured for 22/26 GBCs (proximal dendrites of remaining 4 cells exited image volume), and could reach up to 20 µm (range 3.2-19.6 µm; mean 12.9 (SD 6.2) µm) from the cell body. We used the ten GBCs with complete or nearly complete dendrite segmentations to compute additional summary metrics of dendrite structure. Branches often occurred at near-perpendicular or obtuse angles Nearly all dendritic trees exhibited regions where branches extended alongside one another and could exhibit braiding, whereby branches of the same or different parent branches intertwined, displaying a pattern perhaps unique to mammalian neurons. Dendrites were partitioned qualitatively into categories of little (n=3), moderate (n=4), and dense (n=3) local branching and braiding (*Figure 7A–C*, respectively). EM images reveal the complexity of braided branches and frequent direct contact between them (*Figure 7D–F1*).

Proximal dendrites expanded into a structure from which at least 2 and up to 14 branches extend (7.0 SD (3.8), n=10). We name these structures hubs, due to their high node connectivity (7 branches visible in *Figure 7G*). Secondary hubs were positioned throughout the dendritic tree (*Figure 7H*). One-half (11/22 GBCs) of primary, and some secondary hubs contained a core of filaments that extended through the middle of the structure. This filamentous core was in contact with multiple mitochondria oriented along its axis (*Figure 7J*; and *Figure 7—figure supplement 1*), and was also found in a thickened region of a second order dendrite of one of the two large MCs. Dendrites, as noted previously, have many swellings (*Figure 7H*) along higher order branches. Swellings were more numerous than hubs (range 51–126, mean = 74.9 (SD 26.8)), and were not correlated with the number of hubs ($r^2 < 0.001$; *Figure 7H*). In rank order, dendrite surface area was comprised of dendritic shafts (58%), swellings (28%), hubs (10%), and the proximal dendrite (4%; *Figure 7K*).

## A complete map of synaptic inputs reveals dendrite branches that lack innervation

We report here the first map for locations of all synaptic terminals onto soma, dendrites and axon of a GBC (GBC09; *Figure 8A and B*). In addition to 8 endbulb inputs from ANFs, 97 small terminals contacted the cell body. Together these inputs covered 83% of its somatic surface (*Figure 8C and D*). This neuron had 224 inputs across all dendritic compartments (shaft, swelling, hub, proximal dendrite) (*Figure 8H*). Dendritic and small somatic terminals were typically bouton-sized, contained one or two synaptic sites, and could be linked by small caliber axonal segments to other small terminals across the dendrite and/or soma (*Figure 8A*; cyan arrowheads in *Figure 8A and C*). Previous investigation suggested swellings as preferred sites for innervation (*Ostapoff and Morest, 1991*). However, in our reconstruction, innervation density was similar across most compartments (hubs, 10.4/100 µm²; swellings, 9.3/100 µm²; shafts 9.1/100 µm²), and greatest on the proximal dendrite (24/100 µm²; *Figure 8A, E, G and H*). At least one endbulb (typically 1 but up to 3) on nearly all GBCs (20/21) extended onto the proximal dendrite (mean = 14.5% of endbulb ASA; black arrow in *Figure 8A*). Two endbulbs extended onto axonal compartments of GBC09, indicating that this cell is not exceptional. Somatic endbulbs infrequently (8/159 terminals) innervated an adjacent dendrite of a different GBC.

Notably, entire dendrite branches could be devoid of innervation (black arrows in *Figure 8B*), and instead were wrapped by glial cells, or extended into bundles of myelinated axons (*Figure 8F*). Even though they are not innervated, these branches will affect the passive electrical properties of the cell by adding surface area. We inquired whether these dendrites constitute sufficient surface area and are strategically located to affect excitability of the cell, by generating a model of GBC09 with the non-innervated dendrites pruned. Pruning increased the input resistance from 20.2 to 25.1 MΩ, (*Figure 8I and J*) and increased the time constant from 1.47 ms to 1.65 ms. The threshold for action potential generation for short current pulses decreased from 0.439 to 0.348 nA (*Figure 8J*), but the cell maintained its phasic firing pattern to current pulses (*Figure 8I* compared to *Figure 4—figure supplement 3*, "Half-active"). These seemingly subtle changes in biophysical parameters increased the efficacy for the four largest inputs (0.689–0.786 (14%); 0.136–0.431 (216%); 0.021–0.175 (733%);,

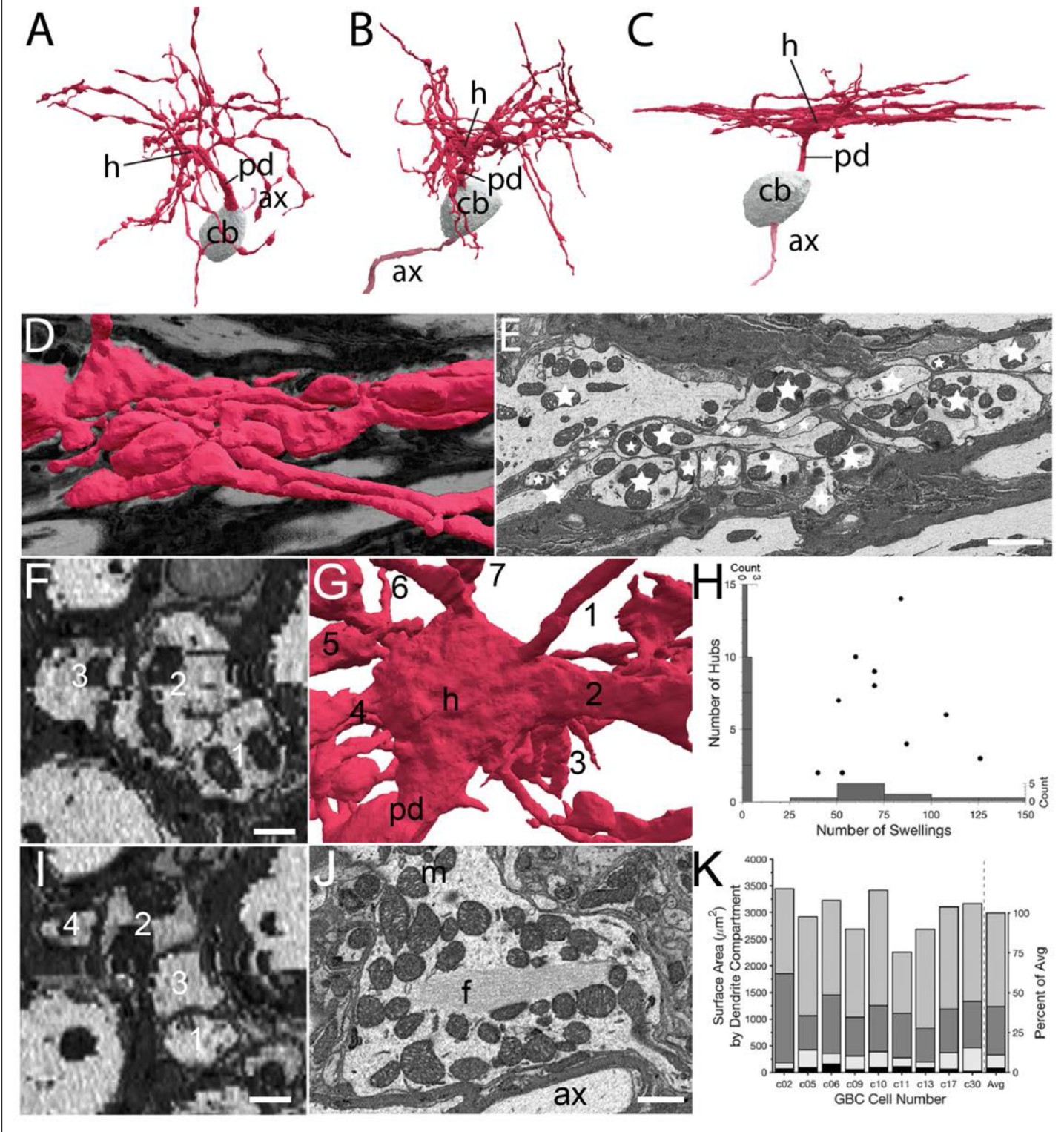

**Figure 7.** Volume EM reveals unique dendrite hub structures and branching patterns. (**A–C**). Dendrites vary in density of local branching and braiding of branches from the same cell, exhibiting (**A**) little, (**B**) medium or (**C**) dense branching and braiding. (**D–E**). Tangential view of dense braiding, showing (**D**) reconstruction of multiple branches in contact with one another and (**E**) a single EM cross-section illustrating contact among the multitude of branches (individual branches identified with stars). (**F, I**). Two locations of cross-cut braided dendrites showing intertwining as change in location of branches (numbers) along the length of the braid. Images are lower resolution because viewing perspective is rotated 90 from image plane. (**G**). Reconstruction of dendrite hub (**h**) and its multiple branches (7 are visible and numbered in this image). (**H**). Swellings and hubs are prominent

*Figure 7 continued on next page*

*Figure 7 continued*

features of GBC dendrites. Histograms of numbers of swellings and hubs plotted along abscissa and ordinate, respectively. (**J**). Core of many hubs is defined by a network of filaments (**f**); also see *Figure 7—figure supplement 1*. Many mitochondria are found in hubs and can be in apparent contact with the filament network. (**K**). Partitioning of dendrite surface area reveals that proximal dendrite (black), hub (light grey), swelling (dark grey) and shaft (medium grey) compartments, in increasing order, contribute to the total surface area for each cell. Averaged values indicated in stacked histogram, to right of vertical dashed line, as percent of total surface area ((right ordinate), and aligned with mean sizes on left ordinate). Scale bars: E, 2 microns; F, I, 0.5 microns; J, 1 micron.

The online version of this article includes the following figure supplement(s) for figure 7:

**Figure supplement 1.** Dendritic Hubs, related to *Figure 7G and J*.

---

0.00092–0.00893 (871%); *Figure 8K and L*). Note that the increase was fractionally larger for the second and third largest inputs compared to the first, reflecting a ceiling effect for the largest input. We also examined how pruning non-innervated dendrites is predicted to affect phase locking to SAM tones (*Figure 8M*). Pruning decreased VS at 100 Hz, thereby sharpening tuning to 200 Hz relative to ANFs. The rMTF (*Figure 8M*, inset) shows a slightly higher rate after pruning of uninnervated dendrites. From these simulations, we hypothesize that GBCs can tune their excitability with functionally significant consequences by extension and retraction of dendritic branches, independent of changes in their synaptic map.

## Discussion

### Volume EM provides direct answers to longstanding questions

Key questions about ANF projections onto GBCs have persisted since the first descriptions of multiple large terminals contacting their cell bodies (*Lorente de Nó, 1933*; *Cajal, 1971*). Volume EM offers solutions to fundamental questions about network connectivity not accessible by LM, by revealing in unbiased sampling all cells and their intracellular structures, including sites of chemical synaptic transmission (for reviews, see *Briggman and Bock, 2012*; *Abbott et al., 2020*). By acquiring nearly 2,000 serial sections and visualizing a volume of over 100 μm in each dimension, we provided reconstruction of the largest number of GBCs to date, permitting more detailed analysis than was possible with previous EM methods that subsampled tissue regions using serial sections (*Nicol and Walmsley, 2002*; *Spirou et al., 2008*; *Ostapoff and Morest, 1991*). Here, we report on a population of GBCs in the auditory nerve root with eccentric, non-indented nuclei, ER partially encircling the nucleus, and somatic contact by a large number (5-12) of endbulbs of mostly smaller size. These cytological features, except for ER patterns, define a subpopulation of GBCs in mice more similar to globular (G) BCs than spherical (S)BCs as defined in larger mammals (*Cant and Morest, 1979b*; *Cant and Morest, 1979a*; *Tolbert et al., 1982*; *Osen, 1969*; *Hackney et al., 1990*) and are also consistent with criteria based on a larger number of endbulb inputs onto GBCs (*Lauer et al., 2013*) than BCs located in the rostral AVCN of rat (likely spherical bushy cells; see *Nicol and Walmsley, 2002*). In cat, the number of endbulb inputs onto GBCs is also large (*Spirou et al., 2005*, mean 22.9) and exceeds the number onto spherical bushy cells (*Ryugo and Sento, 1991*, typically 2).

Nanoscale (EM-based) connectomic studies are providing increasingly large volumetric reconstruction of neurons and their connectivity (*Bae et al., 2021*; *Scheffer et al., 2020*; *Witvliet et al., 2021*). In this report, we add pipelines from neuron reconstruction to biophysically-inspired compartmental models of multiple cells. These models expand on previous GBC models that used qualitative arguments, or single or double (soma, dendrite) compartments (*Joris et al., 1994a*; *Joris et al., 1994b*; *Rothman et al., 1993*; *Rothman and Manis, 2003c*; *Spirou et al., 2005*; *Koert and Kuenzel, 2021*). By matching inputs to a cochlear model (*Zilany et al., 2014*; *Rudnicki et al., 2015*), we created a well-constrained data exploration framework that expands on previous work (*Manis and Campagnola, 2018*). We propose that generation of compartmental models, from high-resolution images, for multiple cells within a neuron class is an essential step to understand neural circuit function. This approach also reveals that there are additional critical parameters, such as ion channel densities in non-somatic cellular compartments, including non-innervated dendrites, that need to be measured. From these detailed models, more accurate reduced models that capture the natural biological variability within a cell-type can be generated for efficient exploration of large-scale population coding.

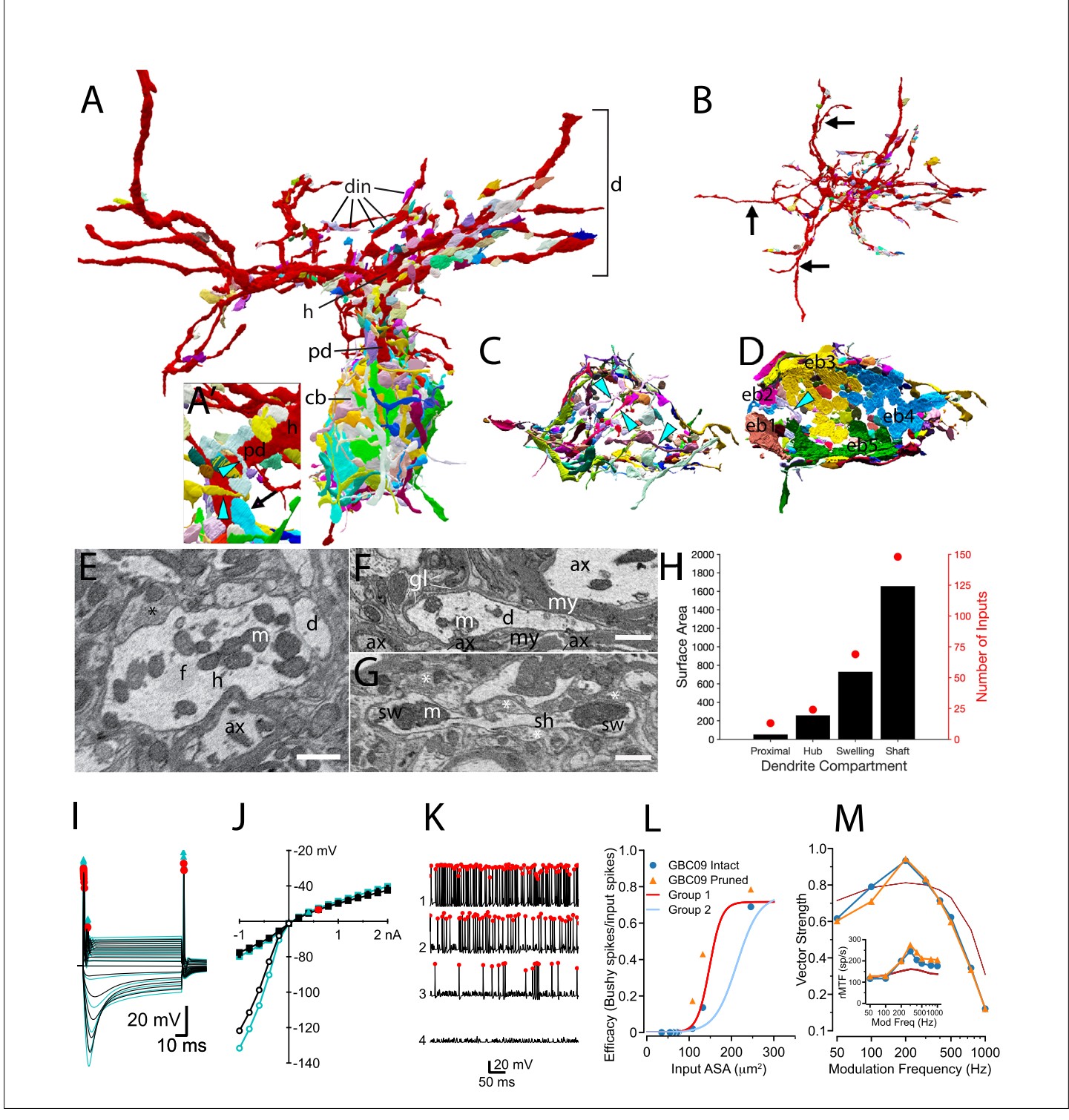

**Figure 8.** Synaptic map of GBC with modeled effects of removing non-innervated dendrites. (**A**) GBC09 oriented to show inputs (din) to dendrites (red, **d**), including proximal dendrite (pd), primary hub (**h**) and cell body (gray, cb). Nerve terminals are colored randomly. Terminals contacting dendrites at higher order sites than the primary hub are bouton-type of varying volume. (**A'**) Closeup view of pd reveals high density innervation by primarily bouton terminals that can be linked by small connections (cyan arrowheads), and extension of a somatic endbulb onto the basal dendrite (arrow). (**B**) Top-down view of dendrites only, illustrating that some branches are not innervated (longest non-innervated branches indicated by arrows) and that other branches are innervated at varying density. (**C**) Bouton terminals innervate all regions of the cb surface. Some boutons are linked by narrow connectors (cyan arrowheads). The cb is removed to better reveal circumferential innervation. (**D**) Inside-out view of cb innervation by endbulbs (ebs; each is numbered and a different color) reveals that they cover most of the cb surface. Cb removed to reveal synaptic face of ebs. (**E**) Cross section through primary hub

*Figure 8 continued on next page*

*Figure 8 continued*

(**h**), showing filamentous core (**f**), mitochondria (**m**), input terminals (asterisks), and contact with dendrite of another cell. (**F**) Non-innervated dendrites (**d**) can be embedded in bundles of myelinated (my) axons (ax), and also ensheathed by glial cells (gl) and their processes (lines). (**G**) Both dendrite swellings (sw) and shafts (sh) can be innervated (asterisks). (**H**) Proximal dendrites are innervated at highest density (number of inputs / surface area), and hubs, swellings and shafts are innervated at similar density. Scale bars: 1 µm in each panel. (**I–M**) Simulation results after pruning the non-innervated dendrites from this cell. (**I**) Voltage responses to current pulses, as in *Figure 4—figure supplement 3*, comparing the intact cell (black traces) with one in which non-innervated have been pruned (cyan traces). (**J**) IV relationship of data in (**I**) Cyan triangle indicates the spike threshold with the dendrites pruned compared to the intact cell (red circle). (**K**) Spikes elicited by the 4 largest individual inputs at 30 dB SPL with the dendrites pruned (compare to data shown in *Figure 4A3*). (**L**) Comparison of the efficacy of individual inputs between intact and pruned cell as a function of ASA. The red and light blue lines (Group1 and Group2) are reproduced from *Figure 4D*. (**M**) Comparison of VS to SAM tones in the intact and pruned configuration. Inset: Rate modulation transfer function (rMTF) comparing intact and pruned dendritic trees. Colors and symbols match legend in (**L**). Dark red line is the rMTF for the auditory nerve input.

The online version of this article includes the following video for figure 8:

**Figure 8—video 1.** Exploration of a globular bushy cell (GBC) and all of its synaptic inputs.

https://elifesciences.org/articles/83393/figures#fig8video1

## Toward a complete computational model for globular bushy cells: strengths and limitations

We propose that the pipeline from detailed cellular structure to compartmental model, informed by physiological and biophysical data on GBCs, provides a framework to highlight missing information that is needed to better understand the mechanisms GBCs employ to process sound, and thereby provide a guide for future experimentation. Some of the information that is missing is inherent in the limitations of the methods employed, and other information must derive from experiments using other techniques.

SBEM has provided an unprecedented spatial scale (a cube of roughly 100 µm per side) for high-resolution reconstruction of entire cells (10 complete, 16 partial) in this brain region. A range of dendrite geometries in terms of branching density are revealed, but the number of reconstructed cells remains constrained by the imaged volume due to the tradeoff between spatial resolution, size of the volume, and time to acquire the images. Although many details of GBC dendrite structure are revealed for the first time, it is not clear whether the full diversity of dendrite structure has been captured. The imaging parameters for this volume were set to permit identification of vesicles, vesicle clusters and synapses, but did not allow us to assess vesicle shape. Thus, the excitatory or inhibitory nature of synapses based on vesicle morphology following glutaraldehyde fixation (*Uchizono, 1965*; *Bodian, 1970*) could not be made. Endbulb neurotransmitter phenotype was known by tracing nerve terminals back to their ANF of origin. The axons of small terminals were not reconstructed, except for selected examples locally. Future analysis of the image volume will require reconstructing longer sections of these axons to reveal regional branching patterns. These patterns can also be matched to other experiments in which axons innervating GBCs from identified source neurons are labeled using genetically driven electron dense markers (*Lam et al., 2015*), and images are collected at higher spatial resolution to permit accurate quantification of synaptic vesicle size, density and shape.

The modeling framework is constrained by anatomical metrics and measurements of biophysical parameters of GBCs from the literature, stemming primarily from brain slice and acute isolated cell experiments. It is encouraging that the response of the model to standard manipulations, such as injection of current steps and activation by tones, illustrates that the fundamental features of the model, including PSTH shapes and firing regularity, align with experimental biology. The purpose of engaging the modeling pipeline, however, is both to identify its limitations, thus revealing key parameters to guide design of future experiments, and also to predict responses of GBCs that can be tested in future in vivo recordings. Given the relatively large number of endbulbs per cell (5-12), it is likely that cells are innervated by ANFs with different distributions of spontaneous rates, and the particular patterns of convergence are expected to affect model responses. Currently, we are not able to assign endbulb size, morphology or axon branching patterns to spontaneous rate classes, although some evidence supports such a correlation (*Wang et al., 2021*; *Sento and Ryugo, 1989*; *Liberman, 1991*; *Rouiller et al., 1986*). Future experiments that define terminal shapes associated with spontaneous rate, perhaps capitalizing on correlations with gene or protein expression (*Sun et al., 2018*; *Shrestha et al., 2018*; *Petitpré et al., 2018*), can be mapped onto this data set. Although synaptic

sites operated independently in the model, the measured nearly constant density of synapses across differing terminal sizes yielded a monotonic relationship between vesicle release and terminal size. The similar mean amplitudes for mEPSCs across experimental recordings from mouse GBCs (*Gardner et al., 1999*; *Wang and Manis, 2005*; *Cao and Oertel, 2010*) argues that parameters such as the number of postsynaptic receptors or synaptic vesicle volume, which could affect synaptic weight, vary similarly across endbulbs and also support a monotonic relationship between weight and endbulb size. Additional factors, such as temporal dynamics of release probability that may differ with size or SR category, or postsynaptic receptor density, can modify this relationship and can be addressed in the model with new experimental data. Furthermore, just as volume EM reveals non-canonical dendrite structures (hubs) and branching properties, the complement of conductances in GBC dendrites, and potential differences among hub, swelling, shaft, proximal dendrite and non-innervated regions is not known. The compartmental models will be improved by new experiments that directly measure these missing conductances and, for all cellular compartments, the co-variance of conductance values for individual cells. Because the models have high spatial resolution, new data can be readily associated with dendrite compartments (proximal dendrite, hubs, swellings, shafts), soma and AIS.

We showed how tuning to SAM tones is predicted to vary based on the entire complement of endbulb sizes onto individual GBCs, but there are few equivalent experimental observations for comparison. The few studies that characterized GBC responses to sound in mice have used limited sets of stimuli (*Roos and May, 2012*; *Kopp-Scheinpflug et al., 2003*; *Willott et al., 1984*) and have not yet provided the kind of structure-function correlations that are available from other species. The only published data that we are aware of for responses to SAM stimuli from mouse CN (*Kopp-Scheinpflug et al., 2003*) show lower VS than our model predicts. However, a direct comparison is difficult because that study reported responses generally for VCN (not specified by cell type), and stimuli were delivered at a high intensity, 80 dB SPL, whereas we used a low-intensity sound that results in maximal SAM VS in low-threshold ANFs. In other species (cat, gerbil, guinea pig) SAM VS is lower in all SR classes of ANFs at intensities well above their thresholds, including at 80 dB SPL, than nearer threshold (*Smith and Brachman, 1980*; *Joris and Yin, 1992*; *Cooper et al., 1993*; *Dreyer and Delgutte, 2006*). This intensity-dependent pattern is also characteristic of neurons in the VCN in gerbil and cat (*Frisina et al., 1990*; *Rhode and Greenberg, 1994*). Thus, our predicted responses to SAM tones are qualitatively consistent with existing experimental data but this conclusion needs to be experimentally tested.

Other future enhancements to the models, by characterizing inputs by their putative excitatory or inhibitory function based on vesicle shape, are an important next step in the evolution of these detailed models. In addition, mapping local and feedback excitatory and inhibitory pathways near CF from specified cellular sources (*Caspary et al., 1994*; *Campagnola and Manis, 2014*; *Xie and Manis, 2013a*; *Cant and Morest, 1978*; *Ngodup et al., 2020*), and knowing their responses to SAM sounds (e.g., for dorsal cochlear nucleus tuberculoventral cells), can help to incorporate their important roles in spectral and temporal processing of GBCs (*Caspary et al., 1994*; *Gai and Carney, 2008*; *Keine and Rübsamen, 2015*; *Keine et al., 2016*). Lastly, we do not have a good handle on the variability of responses within the GBC class that could be used, even in a statistical sense, to constrain model parameters for specific exploration. Given the increasing prevalence of mice in hearing research, especially in studies of cochlear function and pathology, we expect that these data will be forthcoming.

An optimal dataset to test our predictions would match individual cell responses to sound with the detailed structural information from volume EM. Previous connectomic studies that mapped neural activity from cell populations into the EM volume from the same animal used Ca$^{2+}$ imaging to measure spike-evoked activity (*Bock et al., 2011*; *Turner et al., 2022*; *Bae et al., 2021*; *Ding et al., 2023*). However, the resolution of the questions regarding GBC function require near-microsecond precision measurements of action potential timing, and bulk Ca$^{2+}$ signals are too slow to provide this information. Emerging technologies such as genetically encoded voltage-sensitive optical indicators measured with high-speed imaging (*Villette et al., 2019*) may become applicable to this system in future experiments.

The anatomically and functionally constrained model developed here can serve as templates onto which new data are mapped in order to explore in silico representations of GBC function in hearing. The models focus attention on experimental data that is missing in the literature, and become a guide to future studies. Furthermore, because EM reveals subcellular and non-neuronal structures,

this dataset also is branch point for complementary modeling frameworks to understand other cell functions that contribute to the neural encoding of sound.

## Multiple cellular mechanisms to tune excitability

The variability of responsiveness in cells and patterns of convergence in circuits are essential factors that help optimize the representation of sensory information (*Ashida et al., 2019*; *Perez-Nieves et al., 2021*). We predict that dendrite surface area varies sufficiently to adjust spike threshold across the GBC population. Dendrite surface area defined two GBC populations, where cells with smaller areas exhibited greater excitability. Reconstruction of additional cells will be needed to clarify whether excitability is clustered or occurs along a continuum.These two populations did not respect GBC grouping based on the profile of endbulb sizes (coincidence-detection or mixed-mode) or the density of local dendrite branching. Gene expression profiling in mice has revealed differences between BCs in the rostral VCN and caudal AVCN/rostral PVCN (*Jing et al., 2023*), coupled with differences in electrical excitability. Future experiments that combine techniques will be required to relate these molecular profiles to dendrite branching, dendrite surface area, and somatic innervation profiles revealed only by high-resolution structural imaging. Our demonstration of the lack of synaptic innervation along entire branches and increased excitability following their removal, offers an additional mechanism to tune excitability. Although GBCs lack dendritic spines, they may grow or prune dendrite branches in response to cochlear pathology or changing acoustic environment, as has been shown for other brain regions in pathological states (*Furusawa and Emoto, 2020*), experience-driven paradigms (*Berry and Nedivi, 2016*), or during physiological cycles such as estrous or hibernation (*Ferri and Flanagan-Cato, 2012*; *von der Ohe et al., 2006*). The dynamics of dendrite branch remodeling have not, to our knowledge, been examined at high temporal resolution, but are amenable to modern imaging methods such as have been applied to studies of dendritic spine structural plasticity.

We also found that the length of the AIS, which is the spike initiation zone for most neurons (*Bender and Trussell, 2012*), varied across GBCs by 50% (14–21 µm). Changing AIS length, while assuming a constant density of Na$^+$ channels, is predicted to non-linearly change rheobase by 50% (*Figure 5K*). Interestingly, the AIS of each GBC is contacted by multiple small inputs. Inhibitory inputs onto the AIS of other neuron types have been shown experimentally and computationally to modulate spike generation (*Bae et al., 2021*; *Schneider-Mizell et al., 2021*; *Veres et al., 2014*; *Franken et al., 2021*). We reveal that in nearly all GBCs one of the large somatic inputs extends onto the hillock and AIS. In our models, the proximal axon is electrotonically close to the somatic compartment, so further investigation is required to determine whether direct AIS innervation can increase synaptic efficacy for driving spikes. The AIS length and location of Na$^+$ channels have been also shown to be sensitive to the history of neural activity (*Kuba et al., 2010*; *Kuba, 2012*; *Grubb and Burrone, 2010*), and merit investigation in GBCs.

Dendrite surface area and AIS geometry and innervation emerge as potential homeostatic mechanisms to regulate excitability. We expect that reconstructions of a larger population of GBCs will better reveal the distribution of these morphological features, and may clarify additional regulatory mechanisms. Thus, the combination of high-resolution structural analysis and compartmental modeling specifies focused topics for further study.

## Convergence of weak and strong inputs regulates temporal fidelity

We provide the first complete catalogue of numbers of ANF inputs and their sizes (38-270 µm$^2$), revealing a broad range of subthreshold endbulb sizes and raising questions about the functions of smaller endbulbs. GBCs were proposed to achieve their highest temporal fidelity by acting as a coincidence detector for convergence of subthreshold endbulb inputs (*Rothman et al., 1993*; *Rothman and Young, 1996*; *Joris et al., 1994a*). In the present simulations, we took advantage of the ability to selectively activate or silence specific inputs, which allowed us to separately assess the contribution of suprathreshold and subthreshold inputs across a biologically relevant range of strengths. Our simulations predict that only about one-half of GBCs in mice operate strictly in the coincidence detection mode, whereas the remainder operate in a mixed integration mode. A larger sample of cells may clarify whether the sizes of the largest inputs across the population of GBCs are truly a continuum or occur in discrete groupings. Furthermore, we find that by conventional measures of phase locking to an amplitude-modulated tone, the activity of the weaker inputs substantially improves temporal

precision relative to individual ANFs for modulation frequencies up to 200 Hz. In contrast, the largest inputs alone provide better temporal precision than combined inputs only at high modulation frequencies, especially if they are suprathreshold. Supporting the generality of these observations across stimuli, improved temporal precision in the coincidence and mixed modes is also mirrored when using a different measure, the shuffled correlation index, for transient stimuli. Our results are also consistent with simulations showing that small ANF synapses on dendrites can improve temporal precision in the presence of large somatic inputs (*Koert and Kuenzel, 2021*). We also observed that otherwise subthreshold, but large, inputs can effectively drive more spikes by depending on near-simultaneous activation of weaker inputs, than can larger suprathreshold inputs. The suprathreshold input in mixed mode cells decreased VS to ANF values at low frequencies, raising questions regarding their functional contribution to GBC sound encoding. On the other hand, their activation also increased the AP rate, and thereby elevated the rMTF above ANF values at these same frequencies. Thus, we predict that the pattern of convergence of ANF inputs with a wide range of strengths provides a mechanism for improved temporal precision and higher spike rates over part of the range of behaviorally relevant envelope modulation frequencies.

### New dendrite structures

Our high-resolution images revealed a previously undescribed dendrite structure, which we name a hub. The high branching order of hubs helps explain why GBC dendrites are contained locally to the cell body. We also revealed that dendrites branch and align adjacent to one another. This arrangement increases the surface area to volume ratio, which affects the excitability of the cell. Both of these features likely function in part to shorten the overall dendrite electrotonic length and increase the importance of the dendrites in the integration of somatic synaptic inputs. Inspection of published GBC images based on Golgi or tract tracing techniques reveals cells with thickened proximal dendrites (*Webster and Trune, 1982*; *Lorente de Nó, 1981*; *Brawer et al., 1974*). We suggest that some of these represent unresolved dense local branching and hub structures that are better revealed by EM across many sections. We noted that swellings were a prevalent feature of the dendrites and, contrary to reports in cat based on subsampling (*Ostapoff and Morest, 1991*), swellings were innervated at similar densities to shafts. The partition of dendrite compartments into hubs, swellings and shafts may have functional significance if, for example, these structures have differential sources of innervation or are endowed with different densities of ion channels or pumps (*Brownell and Manis, 2014*). The latter may relate to filament bundles and concentrations of mitochondria inside of hubs.

Although our SBEM volumes lacked resolution to assess vesicle shape, it is likely that some of the smaller dendritic inputs are inhibitory (*Gómez-Nieto and Rubio, 2009*). Hubs may also provide efficient sites to nullify excitatory inputs occurring along multiple distal branches through current shunting. Many of the dendritic inputs were linked by short branches. Thus, non-innervated dendrites also afford locations for adaptive regulation of synaptic efficacy via formation or retraction of short branches and new terminals.

## Materials and methods

**Key resources table**

| Reagent type (species) or resource | Designation | Source or reference | Identifiers | Additional information |
|---|---|---|---|---|
| Strain, strain background (Mouse, male) | FVB/NJ | Jackson Laboratory | RRID:IMSR_JAX:001800 | JAX Stock # 001800 |
| Chemical compound, drug | 2,2,2 Tribromoethanol | TCI Chemicals | T1420 | |
| Chemical compound, drug | tert-Amyl Alcohol | TCI Chemicals | P0059 | |
| Chemical compound, drug | xylocaine | Sigma | PHR1257 | |

*Continued on next page*

*Continued*

| Reagent type (species) or resource | Designation | Source or reference | Identifiers | Additional information |
|---|---|---|---|---|
| Chemical compound, drug | heparin | Sigma | H5515 | |
| Chemical compound, drug | Cacodylic acid | EM Sciences | RT12201 | |
| Chemical compound, drug | glutarldeyhde | EM Sciences | 100503–972 | |
| Chemical compound, drug | Paraformaldehyde - EM grade | source | RT19208 | |
| Chemical compound, drug | calcium chloride | Sigma | 223506 | |
| Chemical compound, drug | potassium ferrocyanide | EM Sciences | RT20150 | |
| Chemical compound, drug | Nanopure water | Barnstead International | D11901 | |
| Chemical compound, drug | osmium tetroxide | EM Sciences | 19132 | |
| Chemical compound, drug | thiocarbohydrazide | EM Sciences | 21900 | |
| Chemical compound, drug | uranyl acetate | EM Sciences | 22400 | |
| Chemical compound, drug | lead nitrate | EM Sciences | 17900 | |
| Chemical compound, drug | ethanol | Fisher Chemical | A962P-4 | |
| Chemical compound, drug | acetone | Fisher Chemical | A18-4 | |
| Chemical compound, drug | Gold/palladium sputter target | Ted Pella | 91651 | |
| Chemical compound, drug | Durcopan resin | EM Sciences | 14040 | |
| Chemical compound, drug | Aclar strips | EM Sciences | 50425–10 | |
| Chemical compound, drug | Silver paint | Ted Pella | 16031 | |
| Software, algorithm | Seg3D | The NIH/NIGMS Center for Integrative Biomedical Computing | RRID:SCR_002552 | https://www.seg3d.org |
| Software, algorithm | Blender 2.9 | The Blender Foundation | RRID:SCR_008606 | https://www.blender.org |
| Software, algorithm | syGlass 1.7 | IstoVisio, Inc | RRID:SCR_017961 | https://www.syglass.io |
| Software, algorithm | nrrd_tools | https://digitalcommons.usf.edu/etd/9543 | None | https://github.com/MCKersting12/nrrd_tools |
| Software, algorithm | NEURON V7.7-V8.0 | DOI:10.1017/CBO9780511541612 | RRID:SCR_005393 | http://www.neuron.yale.edu |
| Software, algorithm | Python V3.7–3.10 | Python Software Foundation | RRID:SCR_008394 | https://www.python.org |
| Software, algorithm | cnmodel | PMID:29331233 | None | https://github.com/cnmodel |
| Software, algorithm | Prism V9.3 | GraphPad, Inc | RRID:SCR_002798 | https://www.graphpad.com |
| Software, algorithm | MATLAB R2022a | MathWorks, Inc | RRID:SCR_001622 | https://www.mathworks.com |

*Continued*

| Reagent type (species) or resource | Designation | Source or reference | Identifiers | Additional information |
|---|---|---|---|---|
| Software, algorithm | Adobe Illustrator V26.0.3 | Adobe, Inc | RRID:SCR_010279 | https://www.adobe.com/ products/ illustrator.html |
| Other | Merlin Scanning Electron Microscope | Zeiss Group, Oberkochen, Germany | None | https://www.zeiss.com |
| Other | National Center for Microscopy and Imaging Research | University of California at San Diego | RRID:SCR_016627 | https://ncmir.ucsd.edu |

## Serial block-face scanning electron microscopy

All reagents for transcardial perfusion were purchased from Sigma-Aldrich, unless otherwise noted. An adult male (P60) FVB/NJ mouse (NCI: Frederick, MD and Jackson Laboratory: Bar Harbor, ME) was anesthetized using Avertin (20 mg/kg) injection IP, and perfused transcardially with normal Ringers solution containing xylocaine (0.2 mg/ml) and heparin (20 U/ml) for 2 min at 35 °C followed by 0.15 M cacodylate buffer containing 2.5% glutaraldehyde (Polysciences), 2% paraformaldehyde (Fisher Scientific) and 2 mM calcium chloride at 35 °C for 5 min. The skull was placed on ice for 2 hr, then the brain was removed from the skull and post-fixed for an additional 18 h at 4 °C in the same solution. Brain tissue was cut into 150-µm-thick sections in the coronal plane using a vibratome (Ted Pella) in ice-cold 0.15 M cacodylate buffer containing 2 mM calcium chloride, then washed for 30 min in the same solution. The ventral cochlear nucleus (VCN) was identified in free-floating sections using a stereo-microscope, and sections were photographed before and after dissection of the CN from the surrounding tissue.

The tissue sections were prepared for Serial Block-Face Scanning Electron Microscopy Imaging (SBEM) using an established protocol in our group (*Holcomb et al., 2013*). All staining and embedding chemicals were purchased from EM Sciences unless otherwise indicated, and all water was nanopure filtered (Nanopure Diamond, Barnstead International). Initial staining was performed in a solution combining 3% potassium ferricyanide in 0.3 M cacodylate buffer with 4 mM calcium chloride with an equal volume of 4% aqueous osmium tetroxide, for 1 hr at room temperature (RT). Tissue was processed sequentially through filtered 1% thiocarbohydrazide for 20 min at RT, 2% osmium for 30 min at RT, and 1% uranyl acetate overnight at 4 °C. Tissue underwent triple rinses in $H_2O$ for 5 min each between each step and was triple rinsed in $H_2O$ at RT for 30 min after the final step. Sections were placed into filtered lead aspartate solution (0.066 g lead nitrate dissolved in 10 ml of 0.003 M aspartic acid solution, pH adjusted to 5.5 with 1 N KOH, warmed in a 60 °C oven for 30 min). The tissue was rinsed five times (3 min each), photographed, then dehydrated through graded alcohols into acetone, and flat-embedded in Durcopan resin (Electron Microscopy Sciences) between mylar strips in a 60 °C oven for 48 hr. Tissue samples were again photographed and shipped to the National Center for Microscopy and Imaging Research (University of California San Diego) for imaging.

Resin-embedded tissue was mounted on an aluminum specimen pin (Gatan) using cyanoacrylic glue and precision trimmed with a glass knife to a rectangle ≈0.5-0.75 mm so that tissue was exposed on all four sides. Silver paint (Ted Pella) was applied to electrically ground the edges of the tissue block to the aluminum pin. The entire specimen was then sputter coated with a thin layer of gold/palladium to enhance conductivity. After the block was faced with a 3View ultramicrotome unit (Gatan) to remove the top layer of gold/palladium, the tissue morphology became visible by back-scattered electron detector imaging using a Merlin scanning electron microscope (Carl Zeiss, Inc). A low-magnification image (≈500 X) was collected to identify the proper location in the VCN (caudal and in the auditory nerve root) for serial image collection. This region was selected because it has a high concentration of globular bushy cells (GBC, *Harrison and Irving, 1966*; *Osen, 1969*; *Brawer et al., 1974*). The imaged volume was located at approximately the mid dorsal-ventral location of the VCN. Imaging was performed using a pixel dwell time of 0.5 µs, tissue was sectioned at a thickness of 60 nm, and the imaging run required 7.5 days. Accuracy of section thickness was estimated by assuming circularity of mitochondria and comparing the diameter of longitudinally oriented organelles with diameters measured in the image plane (*Wilke et al., 2013*).

A volume of 148 µm x 158 µm x 111 µm was imaged with an in-plane pixel resolution of 5.5 nm. The image volume contained 31 complete cell bodies, including 26 GBCs. Due to the large size of the

volume (1.4 TB) and the goal of reducing noise in the image, most of the analysis was performed by down-sampling in the image plane. Voxel averaging at 2x2 binning increased the dimensions of each voxel to 11.0nm x 11.0 nm x 60.0 nm. With these imaging parameters, synaptic vesicles can be identified and, in many cases, a post-synaptic density, which appears as darkening on the post-synaptic membrane. Synapses were defined by collections of vesicles near the presynaptic membrane across at least 3 sections and with at least one vesicle in contact with the membrane (*Jackson et al., 2021*). Images were assessed to be of high quality for segmentation due to well preserved membranes, as evidenced also by uniform preservation of tightly wrapped myelin, and the absence of degenerating profiles.

## Segmentation

Seg3D (https://www.sci.utah.edu/cibc-software/seg3d.html, University of Utah, Scientific Computing and Imaging Institute) was used to manually segment the structures of interest from the raw data volume. These structures (somata, nuclei, dendrites, axons, nerve terminals) were identified and segmented according to accepted morphological criteria for the mammalian CNS (*Peters et al., 1991*). The tracing tool was used to paint all pixels interior to the membrane. This strategy permitted the creation of 3D meshes for adjacent structures that did not overlap. Student segmenters were organized into small teams of trained workers supervised by an expert segmenter (who completed a course called Connectomics taught by Dr. Spirou). Expert segmenters reviewed all work by their team of trained segmenters. The 3D meshes of all dendrites were reviewed by expert segmenters and Dr. Spirou in VR (syGlass software; IstoVisio, Inc), overlaid onto the EM image volume so that anomalous branches and structures could be identified, and enclosed ultrastructure and membranes could be incorporated into the evaluation. Tracing the dendrites of all 31 cells provided an internal self reference preventing incorrect assignment of branches to a particular cell. Tracing of dendrites for import into the modeling environment provided additional rigorous review for the subset of 10 cells with complete or near-complete dendritic trees. Endbulb terminals were traced by the same segmenting teams with the same review procedures. Tracing all large inputs and several smaller inputs onto the 21 GBCs reported here also provided an internal check that branches of inputs were not missed or assigned to the incorrect terminal. Testing methods for calculation of the ASA followed by performing the calculation for all large inputs onto all cells provided additional rigorous review of the large terminal segmentations.

Fascicles of nerve fibers traverse the volume in the coronal and sagittal planes. ANFs formed the fascicles in the coronal plane. These fascicles were outlined in every 100th section so they could be tracked to determine their extent of splitting and merging. Branches from axons within the fascicles that led to endbulb terminals were also segmented and tabulated, to determine whether axons in particular fascicles gave rise to endbulb terminals within the volume or tended to converge onto the same cellular targets. Terminal size was quantified by measuring the apposed surface area with the postsynaptic membrane, omitting regions where the membranes were separated by intervening glia or extracellular space. We reconstructed the terminals onto each cell that appeared larger than bouton terminals. On two cells we reconstructed all terminals, and from these data we created a histogram of terminal sizes and a definition of minimum size for the large terminal class. We then verified that terminals larger than this threshold were indeed branches of ANFs (see Results). All endbulb axons were traced visually from the terminal retrogradely to their parent ANF or to the location where they exited the image volume. The axon and fiber diameters were calculated from a subset of fibers that had a segment with a straight trajectory either parallel or perpendicular to the image plane, in order to calculate their axon and fiber diameters. A similar procedure was applied to a subset of ANFs (see *Figure 3F*). To visualize the spatial relationship of endbulbs and ANF branches to ANF fascicles, all these structural elements for all endbulb inputs to four cells were segmented using the tracing tool in syGlass.

## Three-dimensional reconstruction

3D models of the structure of interest were exported from Seg3D as a VTK file and converted to OBJ format using a custom Python script or, in newer versions of the software, exported directly as OBJ files. The meshes in OBJ format were imported into Blender (https://www.blender.org) for processing. Meshes were first decimated by using the decimate modifier tool in collapse mode to

merge neighboring vertices progressively while considering the shape of the mesh (*Low and Tan, 1997*). The meshes are then smoothed using the smooth modifier tool. While these mesh processing steps are suitable for visualization, they do not produce sufficiently accurate surface area or volume measurements. Thus, we evaluated more consistent mesh processing algorithms.

We implemented accurate mesh processing by applying the GAMer2 algorithms and procedures systematically to all meshes in order to create so-called computational meshes (*Lee et al., 2020a*). Surface meshes of segmented objects were generated by performing marching cubes, and produced structures having greater than 1 million vertices due to the high-resolution images and anisotropic sampling during imaging (resolution in x-y plane was ten times resolution in z direction). Anisotropic sampling generates a stair-step effect in the rendering (*Figure 1—figure supplement 1A*). Initial vertex decimation was designed to generate meshes containing 100,000-300,000 vertices and reduced time to perform subsequent processing. Experimentation revealed this size range to be the minimum that preserved geometry upon visual inspection. Next, twenty iterations of angle-weighted smoothing (AWS) were applied, which generated nearly equilateral triangles for the mesh faces (*Figure 1—figure supplement 1B*). This geometry is a characteristic of a well-conditioned mesh, which maintains complete surfaces through subsequent processing (*Shewchuk, 2002*). Two iterations of normal smoothing (NS) were then applied which, in combination with AWS, resulted in a reduction of surface area. The surface area reached an asymptote after the second NS step, confirmed by running three cell bodies through a second round of AWS and NS, indicating that the stair-step effect was minimized after the first round of AWS and NS (*Figure 1—figure supplement 1C*). We visually inspected the meshes during mesh processing and confirmed that all features of the mesh were well-preserved and stair step features were removed after one round of AWS and NS (*Figure 1—figure supplement 1B*). Therefore, we determined this stage of mesh processing to be an accurate stopping point.

## Assignment of synaptic weights

We assigned synaptic weights as a density of synapses per square micron of directly apposed pre- and postsynaptic membrane, the latter of which we term the apposed surface area (ASA). EM affords the opportunity to measure accurately the membrane apposition, and account for features such as extended extracellular space (*Cant and Morest, 1979a*; *Rowland et al., 2000*), where the membranes separate, and interposition of glial processes. We generated an algorithm and custom Python script to identify only the ASA and calculate its summed value for each nerve terminal (https://github.com/MCKersting12/nrrd_tools; *Kersting, 2020*). This script reads the original segmented image volumes of the two objects contacting one another, which may have been traced in different subvolumes of the original volume (subvolumes were created to permit multiple segmenters to work in parallel), and transforms them to have the same origin (pixel-spacing, height, width, and length). If the segmented terminal and postsynaptic cell have overlapping voxels, the overlap is removed from the soma because the terminal segmentations were typically more accurate. Next, the terminal is dilated by 3 voxels in the x-y plane and then, because the volume is anisotropic, another 3 voxels in all directions. The dilation in z was tested and this value was chosen based on visual inspection to provide overlap selectively of the ASA. The overlapping region between the dilated terminal and the soma volume is extracted as a separate volume, and the marching cubes algorithm is performed on this separated volume. The surface area of the resultant mesh, which appears as a flattened volume, is divided by two because we are only interested in the contact area to generate the ASA.

Synapses can be identified in our SBEM volume by clustering of synaptic vesicles along the presynaptic membrane in at least three serial sections, direct contact of at least one vesicle with the presynaptic membrane, and a concavity in the postsynaptic membrane, the latter of which is typical of endbulb terminals in the cochlear nucleus in aldehyde fixed tissue (*Spirou et al., 2008*; *Cant and Morest, 1979a*; *Ryugo et al., 1997*). A postsynaptic density is typically found but is not present in all cases, so was not used as an explicit criterion. Each large input contains multiple synapses, so the number of synapses was quantified for 23 terminals of varying sizes, and density (#synapses/$\mu$m$^2$) was calculated using the ASA for each terminal. The average synapse density was applied to terminals for which the ASA was determined but synapses were not counted, to achieve an estimate of the number of synapses in each terminal reconstructed in this study.

## Model generation

Biophysicallybased models were generated for each reconstructed cell, using the ASA data for individual auditory nerve inputs, and the compartmental reconstructions. The modeling was performed as a predictive exercise, using previously measured biophysical parameters for synapse release dynamics, postsynaptic receptors, and ion channels, along with a standard model of auditory nerve responses to sound. The principal free parameters were the densities of channels in different cell compartments. The channel densities were calculated based on the ratios of densities for somatic models in a previous study (*Rothman and Manis, 2003c*), measured densities in voltage clamp from mouse GBCs for the low-threshold potassium conductance, and relative densities in the axon initial segment and hillock from other central neurons. Because ion channel densities in the dendrites of bushy cells have not been measured, we bracketed the likely range by testing models with passive dendrites, fully active dendrites (densities were the same as in the soma) and half-active dendrites. Thus, the models are predictive given the constraints of unmeasured channel densities. To accomplish this, the models were built up in a series of steps: morphological reconstruction, surface area adjustments, base channel density adjustment, and overall channel density assignment. Synaptic conductances were constrained by previous measurements (*Raman and Trussell, 1992*; *Xie and Manis, 2013b*), and the only free variable was the number of sites for each multi-site synapse, which was set according to the ASA measurements and release site counts from the SBEM material.

## Translating reconstructions to NEURON models

We rendered the SBEM mesh into a modified version of the SWC file format (*Cannon et al., 1998*) using the tracing tool in syGlass. Each reconstructed part of the cell is represented as a series of conical frustums with starting and ending radii. We also annotated groups of points with a named morphological feature of the section. Identified morphological features were given new tags in the SWC file, and included the myelinated axon, axon initial segment, axon hillock, soma, proximal dendrites, dendritic hubs, distal dendrites, and dendritic swellings. Next, the SWC files were translated to HOC files using a Python script. The script added groups of SWC points in a 3D shape format (pt3d) to create short sections composed of at least three and up to 50 segments. This translation retained the detailed geometry of the cells. Comment fields in the HOC files referenced the original SWC point for each 3D point in Neuron, which facilitated mapping voltages in processes back to the original mesh representation, and confirming that the translation proceeded correctly. This annotation also allowed us to perform manipulations that removed specific parts of the original reconstruction.

We then compared the original SBEM mesh surface area representations with those of the 3D geometry HOC files. The mesh represented the cell surface at a high resolution that captured membrane crenelations, even after reducing the mesh density with GAMer2 (*Lee et al., 2020b*) and subsequent smoothing. In contrast, the SWC and HOC representations capture the mesh structure using simple frustrated cones, which have smooth surfaces. Consequently, the mesh surface area was always significantly greater than the surface area computed from the HOC representation. The surface area determines the capacitance and plays a fundamental role in establishing ion channel densities and the transmembrane leak resistance in the model cells. We therefore compensated for these surface area differences by inflating the compartment diameters in the HOC file by the ratio between the mesh and HOC areas, while not changing the lengths. Separate inflation factors were calculated for the soma and for the entirety of the dendritic tree, because the mesh to HOC surface area ratio for these regions were different. NEURON instantiates compartments (as truncated conical segments) from the 3D reconstructions. However, there appears to be no analytical solution to the inverse problem of recalculating the segment areas from the point diameters. Therefore, we computed the inflation factor iteratively by adjusting the diameters until the reconstructed area, as computed from NEURON, matched the mesh area. For the bushy cells, the soma inflation factor averaged 1.486 (SD 0.227), and the factor for the dendritic tree averaged 1.506 (SD 0.145). The ratio of the soma inflation factor to the dendrite inflation factor for different reconstructions varied from 0.78 to 1.38 (mean 0.995, SD 0.195). The last step in establishing the geometry for simulations was determining the number of segments necessary to maintain an appropriate spatial discretization. The number of segments for each section was recomputed using the d-$\lambda$ rule (*Carnevale and Hines, 2006*), at 1000 Hz. Because many of the reconstructions already had short section lengths, this step affected only a fraction of the sections for any given cell. All current clamp simulations were run with a time step of 25 μs.

## Ion channels and receptors

Cells were 'decorated' with Hodgkin-Huxley style ion channels based on biophysical measurements from previous studies. The kinetic measurements for K$^+$channels were obtained from acutely isolated bushy neurons that lacked dendritic trees (*Rothman and Manis, 2003a*), scaled to 37°C (*Rothman and Manis, 2003b*). We drew K$^+$channel density estimates from measurements made from cells in mouse brain slices (*Cao et al., 2007*), scaled as described below. Sodium channels were represented by a modified model (*Xie and Manis, 2013b*), which incorporated cooperative interactions between channels (*Huang et al., 2012*; *Ilin et al., 2013*; *Manis and Campagnola, 2018*). Actual conductance densities for the dendrites, axon hillock, axon initial segment, and nodes of Ranvier are not known. To address these uncertainties, we decorated the cell compartments using density distributions that have been estimated for other neurons, as described next.

### Axons

Axons were reconstructed from the soma to the first internodal (myelinated) region for 8 of the 10 reconstructed bushy cells. Data from mouse bushy cells from *Yang et al., 2016* indicates that the Na$^+$ channel density is lower in the soma than in the axon hillock and that the action potential initiation begins distally, likely in the AIS. Lacking direct measurements in bushy cells, we used the experimental and model data from *Kole et al., 2008* from layer V cortical neurons to guide the relative channel densities. The axon hillock channel density for Na$^+$ channels was set to five times that of the soma, and the initial segment was 100 times that of the soma. The hillock and AIS compartments were each decorated uniformly, to approximate the uniform distribution reported for immunostaining of Na$^+$ channels (*Kuba et al., 2015*), although there is some data suggesting that channel density and composition vary with distance from the soma (*Lorincz and Nusser, 2008*; *Hu et al., 2009*). The assignment of spatially uniform conductance densities to the AIS represents a first-order assumption, as we lack experimental data with appropriate resolution to justify other distributions in GBCs. With this decoration, the total AIS Na$^+$ conductance in the model is a function of AIS length, and therefore also affects action potential threshold and amplitude. Variations in AIS length have been correlated with neuronal excitability (*Grubb and Burrone, 2010*; *Kuba et al., 2010*; *Kim et al., 2019*; *Kaphzan et al., 2011*), and tonotopic position in nucleus laminaris (*Kuba et al., 2006*). Relative Na$^+$, K$^+$channel and I$_h$ channel densities are shown in *Table 1*.

For GBC02 and GBC05, the axon left the tissue block before becoming myelinated. To compensate, we replaced the axon hillock, initial segment and first myelinated region with a standard axon based on the average axon lengths and diameters from the other eight cells for simulations of these cells. These cells were not used in evaluating the effects of AIS length on excitability, although their data is plotted alongside the other cells for comparison.

### Dendrites

Based on the SBEM measurements, the surface area of bushy cell dendrites ranged from 2.43 to 3.23 (mean 2.76 SD 0.24) times the cell body area. Although bushy cell dendrites are short, they have a large diameter and consequently represent a substantial capacitance and conductive electrical load to the soma. The distribution of ion channels on GBC dendrites is not known. Qualitative immunostaining studies hint at the presence of HCN and low-voltage activated K$^+$ channels in at least the proximal GBC dendrites (*Koch et al., 2004*; *Oertel et al., 2008*; *Pál et al., 2005*; *Wang et al., 1993*) (but see *Perney and Kaczmarek, 1997* where dendritic staining for the high-voltage activated channel Kv3.1 is visible in stellate cell dendrites but not clearly visible in bushy cell dendrites in rat). However, with relatively few synaptic inputs and a limited role for active dendritic integration, it seems likely that voltage-gated ion channels may not be present at high densities in the dendrites. To account for

**Table 1.** Densities of channels used to decorate the axon compartments of bushy cells.

Values are given as ratios relative to the standard decoration of the somatic conductances.

Decoration Type

| Channel | Myelinated axon | AIS | AH |
|---------|-----------------|-----|-----|
| *Na* | 0.0 | 100.0 | 5.0 |
| $K_{HT}$ | 0.01 | 2.0 | 1.0 |
| $K_{LT}$ | 0.01 | 1.0 | 1.0 |
| $I_H$ | 0.0 | 0.5 | 0.0 |
| *Leak* | 0.00025 | 1.0 | 1.0 |

the potential roles of dendritic channels, we therefore bracketed the conductance density range with three models. In each of these models, we decorated all types of dendritic compartments (proximal and distal dendrites, dendritic hubs, and dendritic swellings) with the same conductance densities. First, we used a model in which the densities of the channels in the dendrites were half of those in the soma (Half-active). The other two models addressed the extremes of possible channel densities. In the 'Passive dendrite' model, the dendrites were uniformly decorated only with leak channels. In the 'Active dendrite' model, the dendritic channel density was set uniformly to the somatic channel density for all channels. We refer to these models below as the dendritic decoration configurations.

## Conductance Scaling

To properly scale the conductances into the somatic and dendritic compartments, we began with the low-voltage activated channel, $g_{KLT}$, which was measured under voltage clamp to be 80.9 (SE 16.7) nS in CBA mice (*Cao et al., 2007*). Next, to set a baseline value for the conductances, we first computed the mean somatic surface area from the SBEM mesh reconstructions (1352.1 (SD 164.9) $\mu m^2$, N=26 bushy cells), and for dendrites from the ten complete reconstructions (3799.5 (SD 435.8) $\mu m^2$, N=10 bushy cells). We then chose one cell whose somatic and dendritic areas were closest to the mean of these distributions (GBC17: somatic surface area = 1357.6 $\mu m^2$; dendritic 3707.7 $\mu m^2$) to adjust $g_{KLT}$. The use of the 'average' cell for this step was chosen to be consistent with the use of the mean value from *Cao et al., 2007*. We then adjusted $g_{KLT}$ by computing the measured $g_{KLT}$ from a voltage clamp protocol that mimicked experimental measurements (steady-state currents with 100ms pulses) with only $g_{KLT}$ and a leak conductance inserted into the soma and dendrites for each of the three dendritic distribution assumptions. The soma was initially decorated with $g_{KLT}$ channels at a fixed density of 2.769 mS/cm² based on a maximum conductance measured in vitro of 80 nS and a measured cell capacitance of 26 pF (*Cao et al., 2007*). However, this capacitance corresponds to a surface area of 2889 $\mu m^2$, which is more than twice the area of the measured somas, and is also significantly larger than other previously reported values (12 pF in acutely isolated neurons from guinea pig *Rothman and Manis, 2003a*, 9–12 pF in rat pup bushy cells in slices (*Xu-Friedman and Regehr, 2008*), 9–22 pF in adult CBA mouse bushy cells, Xie and Manis, unpublished). To investigate this discrepancy, we measured the input capacitance (as seen by a somatic electrode) using voltage clamp simulations of the reconstructed cells. The voltage-clamp simulations were stepped at 5 $\mu s$ with 1M$\Omega$ of uncompensated series resistance ($R_s$), to approximate the experimental situation that used 90% compensation of ~ 11 M$\Omega$ $R_s$ (*Cao et al., 2007*). Voltage steps from –80 to –90 mV were applied to models with only $g_{KLT}$ and $g_{leak}$ channels in the membrane, which yielded values of 13 pF, based on the fastest membrane charging time constant of ~15 $\mu s$, consistent with the studies cited above. This corresponds to a membrane area of 1460 $\mu m^2$, close to 1358 $\mu m^2$ measured for the soma area of this cell. We then ran additional voltage clamp simulations with steps from –80 to +20 mV to measure $g_{KLT}$. Total $g_{KLT}$ was measured from the V-I relationship by fitting a Boltzmann function to the steady-state portion of the simulated currents (*Figure 4—figure supplement 2*), after correcting the membrane voltage for the drop across the series resistance, $R_s$. We iteratively made a linear prediction after each adjustment, by calculating the ratio between the measured conductance and the target value of 80 nS, and applied this to rescale $g_{KLT}$ across the entire cell, according to the relative values in *Table 2*. Three to five iterations were adequate to arrive within 1% of the target value for $g_{KLT}$ (as measured from the soma) for each of the three dendritic decoration models for the test cell. Once $g_{KLT}$ was determined, the ratio of $g_{KLT}$ to the original model channel density was then calculated, and applied to all of the other channels at the soma, relative to their total cell conductances in the original models. Based on the measurements and models of *Xie and Manis, 2013b* and measurements of *Cao et al., 2007*, the original model conductances were: $g_{KLT}$=80 nS; $g_{Na}$=500 nS, $g_{KHT}$=58 nS, and $g_H$=30 nS.

**Table 2.** Densities of channels in dendrites for three models.

Values are relative to somatic conductance. Leak is in mS/cm².

**Dendrite Decoration Type**

| Channel | Passive | Half-Active | Active |
|---------|---------|-------------|--------|
| *Na* | 0.0 | 0.5 | 1.0 |
| $K_{HT}$ | 0.0 | 0.5 | 1.0 |
| $K_{LT}$ | 0.0 | 0.5 | 1.0 |
| $I_H$ | 0.0 | 0.5 | 1.0 |
| *Leak* | 0.0693 | 0.0693 | 0.1385 |

The dendritic channel densities were then computed relative to the somatic density (except for leak) (*Table 2*). Thus, with this approach, we anchored the model ion channel densities according to our morphological measurements with experimental measurements of $g_{KLT}$ in the same species.

## Auditory nerve inputs

Auditory nerve spike trains were computed using the *cochlea* package (*Rudnicki et al., 2015*), which is a Python wrapper around the widely-used model of *Zilany et al., 2014*. These simulations were incorporated into, and controlled by, *cnmodel* (*Manis and Campagnola, 2018*). Although the spike trains generated by these simulators were based on data from cat ANFs, the responses for mouse auditory nerve are quite similar, including irregular interspike intervals and the thresholds are similar in the central range of mouse hearing (*Taberner and Liberman, 2005*). Tonal acoustic stimuli were generated at 100 kHz with rise-fall times of 2.5ms, and durations from 100 to 1000ms. Clicks were generated as 100 µs pulses. The intensity was expressed in dB re 2×10$^{-5}$ Pa (dB SPL). For tonal stimuli, the frequency was set to 16 kHz to avoid low-frequency phase locking.

For some simulations, single-frequency tones at 16 kHz were amplitude modulated with a sinusoidal envelope (100% modulation) at frequencies between 50 and 1000 Hz. The depth of response modulation in ANFs is critically dependent on the average stimulus intensity as well as ANF SR (*Smith and Brachman, 1980*; *Joris and Yin, 1992*; *Joris et al., 2004*; *Wang and Sachs, 1993*) and this sensitivity continues to be evident in cochlear nucleus neurons (*Moller, 1972*; *Frisina et al., 1990*; *Wang and Sachs, 1994*). We tested responses of the GBC models to SAM tones at an intensity that produces the highest synchronization in the high-spontaneous ANFs, 15 dB SPL, as well as at 30 dB SPL (see *Figure 7—figure supplement 1* for the VS as a function of level in the ANF model). Testing was performed with only high-SR ANFs as inputs, consistent with observations in cats that GBCs are principally innervated by high-SR inputs (*Liberman, 1991*). Testing by including other SR groups would be expected to show higher synchronization at high sound levels (*Wang and Sachs, 1994*) as the medium and low SR fibers continue to synchronize to the envelope. However, this would require making specific assumptions about the relationship between ASA and SR in order to appropriately assign SR groups. While recent data (*Wang et al., 2021*) suggests that some mouse GBCs may receive a greater proportion of medium and low-SR inputs than previously suggested for cat, we considered exploration of this dimension in the context of our simulations beyond the goals of the current study.

## Endbulb synapses

The endbulb synapses were modeled using a stochastic multisite release model, as described previously (*Xie and Manis, 2013b*; *Manis and Campagnola, 2018*) and incorporated into *cnmodel*. Briefly, the release at each endbulb terminal is initiated when an action potential is generated by the auditory nerve model. Each synapse in the terminal then can release transmitter with a release probability, $P_r$ in the range [0,1]. In the present simulations, the release probability was held fixed over time (it was not a function of the history of release event times). Whether a synapse will release or not is determined by drawing a random number from a uniform distribution, and if the number is less than $P_r$, then a release event is initiated. Transmitter time course was computed by convolution of a Dirac pulse with a bi-exponential function to mimic diffusion across the synaptic cleft, and the concentration time course at the postsynaptic receptors is computed by summing each release event with an ongoing cleft concentration. This glutamate transient then drives postsynaptic receptors. The postsynaptic receptors are based on fast AMPA receptors at the endbulbs in the nucleus magnocellularis of chicken (*Raman and Trussell, 1992*), with kinetics adjusted to match recorded currents at the mouse endbulb (*Xie and Manis, 2013b*). The AMPA receptor model conductances were also adjusted to match measurements of mEPSCs at mouse bushy cells. The receptor model includes desensitization, and the current through the receptor channels includes rectification of the current-voltage relationship by internal channel block from charged polyamines (*Woodhull, 1973*; *Donevan and Rogawski, 1995*). The cleft glutamate also interacts with NMDA receptors in the synapse, based on the model of *Kampa et al., 2004*. NMDA receptor conductances were scaled to match the to the voltage-clamp measurements in *Cao and Oertel, 2010*. Each release site of the terminal is treated independently, ignoring the possible consequences of transmitter spillover. A time-dependent increase in release latency is observed experimentally (see *Manis and Campagnola, 2018*), but was disabled in the simulations reported here because it has not been fully characterized. The number of synapses at each endbulb

is calculated using the ASA and average synapse density as determined from the SBEM data. For all simulations here, the density was 0.7686 synapses/μm$^2$ .

## Spike detection

Spikes in bushy neurons are often small and of variable amplitude, and the EPSPs can be large (10 s of mV). Simple approaches using a fixed voltage or slope threshold are not reliable for discerning spikes from EPSPs with somatic recordings. We, therefore, used the method of *Hight and Kalluri, 2016* to detect spikes based on the width of the peak and the rising and falling slopes. Spike detection parameters were set exactly as in *Hight and Kalluri, 2016*.

## Cross-correlation

Correlations between postsynaptic spikes and the input spike trains were calculated as cross-correlations against each of the independent inputs to a cell. The correlations were calculated using the 'correlogram' routine from Brian1.4, and were taken with respect to the time of the postsynaptic spike. Presynaptic spikes occuring after the postsynaptic spike are not shown. The result is presented in Hz (spikes/second), as the rate of coincidences between presynaptic spikes from each input and the postsynaptic spike in each time bin, at a time resolution of 0.1ms.

## Rate modulation transfer function

The rate modulation transfer function (rMTF) was calculated as described in *Walton et al., 2002*. The rMTF was calculated as the average rate at each modulation frequency for spikes starting 250ms after stimulus onset and ending at the time corresponding to the starting phase during a 1 s SAM tone. The window for the rate calculation set in this way to be sure that all frequencies included complete modulation cycles.

## Entrainment

Entrainment was calculated from the interspike interval distribution as described in *Joris and Yin, 1992* and *Rudnicki and Hemmert, 2017*, with one modification. At low modulation frequencies (50 and 100 Hz), multiple spikes could occur per modulation cycle, both in the auditory nerve and in the bushy cells. This led to low values of entrainment, even though the cells were firing on most cycles. To minimize this confound, we set the lower bound of included interspike intervals to $0.5/f_{mod}$, rather than 0 (the upper bound remained $1.5/f_{mod}$). This does not entirely eliminate the presence of spontaneous or multiple spikes contributing to the entrainment index at low frequencies, but it reduces the chances that they will be included. The ISI distribution was derived from spikes starting 250ms after tone onset and ending at the longest interval that fell within a complete cycle (determined from the starting phase) during a 1 s SAM tone.

## Spike timing analysis

Vector strength was computed using the standard equations (*Goldberg and Brown, 1969*), using spikes taken from the last 750ms of 100 repetitions of 1 s long SAM stimuli. To estimate the error of the vector strength calculation, vector strength was calculated for 10 groups of 10 consecutive repetitions, and the mean and SD computed. Responses with fewer than 50 spikes were not calculated (this appeared only for GBC10 for the configuration with only the largest input active). Vector strength for ANFs was calculated across all spikes of all ANFs connected to the postsynaptic cell. We also calculated shuffled autocorrelations using the method of *Louage et al., 2004* for both SAM stimuli and click stimuli. These calculations were verified to reproduce Figure 2 of *Louage et al., 2004*.

## Action potential current threshold measurement

The minimum current required to elicit an action potential (rheobase) was measured in response to a brief current pulse (20ms) of variable amplitude. An iterative binary search procedure was used to identify the threshold, with a terminal step size of 1 pA. Ten to twenty iterations were sufficient to resolve threshold to this precision.

## Modeling software environment

The entire set of simulations were controlled and analyzed by additional Python (V3.7.8, 3.8.6, 3.9.1, 3.10.0) scripts (*VCNModel*). *VCNModel* controlled simulations and organized simulation result files,

read cell morphology files into NEURON Carnevale and Hines (2006), and decorated the cells with channels using tools from *cnmodel* (https://www.github.com/cnmodel; *Manis, 2022*). Parametric simulations were managed by shell scripts (bash, zsh) that called the Python scripts. Simulations reported here were run with NEURON 7.7, 7.8.1, 8.0, and 8.1 on an 8-core MacPro (2013), a MacBook Pro (2017), and a 20-core MacStudio (2022); there was no difference in the results of the underlying auditory nerve, bushy cell, or synapse models as determined by the unit tests in *cnmodel* for any versions of NEURON, Python, or hardware. The anatomical structure of the reconstructions was defined by the NEURON HOC files, and the channel densities were set from text (human readable) tables managed by *cnmodel*. The *VCNModel* scripts computed scaling of cell areas (inflation of the SWC/HOC files to match the mesh areas), control of "experiments" (for example, only activating selected AN terminals), data management, plotting, and analysis. Analysis of current voltage relationships and spike detection was handled by the *ephys* package (https://www.github.com/pbmanis/ephys; *Manis, 2023b*). Plots were generated using matplotlib (versions 3.2.0–3.5.2) and seaborn (version 0.11.2).

## Acknowledgements

We thank Dr. Ken Hutson for comments on the manuscript, the many student tracers who contributed to cell segmentation, Thomas Deerinck (UCSD) for development of tissue staining protocols and managing image acquisition, Brian Pope for tissue processing and Lyra Gaboardi for movie production. We also thank the reviewers of this manuscript for their very constructive comments. This work was supported by NIH/NIDCD grant R01 DC015901, "The Nanoscale Connectome of the Cochlear Nucleus" (GAS, PBM, MHE). Development of the modeling platform cnmodel, and extensions that implemented decorations of dendrites and various manipulations of the cells wer supported by NIH/NIDCD grant R01 DC004551, "Cellular Mechanisms of Auditory Information Processing" (PBM). SBEM data acquisition was performed at the National Center for Microscopy and Imaging Research, with support from NIH/NINDS grant U24 NS120055 (MHE). Deposition and management of acquired raw and derived EM data within the Cell Image Library was further supported by NIH/NIGMS grant R01GM82949 (MHE).

## Additional information

### Competing interests

George A Spirou: George A. Spirou is the Chief Scientific Officer of IstoVisio, Inc, which makes the syGlass software used to visualize the EM volume and reconstruct the neurons. The other authors declare that no competing interests exist.

### Funding

| Funder | Grant reference number | Author |
|---|---|---|
| National Institute on Deafness and Other Communication Disorders | R01 DC015901 | George A Spirou |
| National Institute on Deafness and Other Communication Disorders | R01 DC004551 | Paul B Manis |
| National Institute of General Medical Sciences | R01 GM082949 | Mark H Ellisman |
| National Institute of Neurological Disorders and Stroke | U24 NS120055 | Mark H Ellisman |

The funders had no role in study design, data collection and interpretation, or the decision to submit the work for publication.

## Author contributions

George A Spirou, Conceptualization, Software, Formal analysis, Funding acquisition, Investigation, Visualization, Methodology, Writing – original draft, Project administration, Writing – review and editing; Matthew Kersting, Software, Formal analysis, Investigation, Visualization, Methodology, Writing – original draft; Sean Carr, Bayan Razzaq, Carolyna Yamamoto Alves Pinto, Mariah Dawson, Formal analysis, Investigation, Visualization; Mark H Ellisman, Investigation, Visualization, Methodology; Paul B Manis, Conceptualization, Software, Formal analysis, Funding acquisition, Investigation, Visualization, Methodology, Writing – original draft, Writing – review and editing

## Author ORCIDs

George A Spirou ⬤ https://orcid.org/0000-0001-7677-3585
Matthew Kersting ⬤ http://orcid.org/0000-0002-7632-1762
Sean Carr ⬤ http://orcid.org/0000-0002-6757-9104
Bayan Razzaq ⬤ http://orcid.org/0000-0002-1307-8531
Carolyna Yamamoto Alves Pinto ⬤ http://orcid.org/0000-0001-6735-045X
Mariah Dawson ⬤ http://orcid.org/0009-0005-9359-2459
Mark H Ellisman ⬤ http://orcid.org/0000-0001-8893-8455
Paul B Manis ⬤ http://orcid.org/0000-0003-0131-8961

## Ethics

All procedures involving animals were approved by the West Virginia University (WVU) InstitutionalAnimal Care and Use Committee, protocol #15-1201 (G.A. Spirou, PI) and were in accordance with policies of the United States Public Health Service. No animal procedures in this study were performed at other institutions. The perfusion of the mouse was performed under avertin anesthesia.

## Decision letter and Author response

Decision letter https://doi.org/10.7554/eLife.83393.sa1
Author response https://doi.org/10.7554/eLife.83393.sa2

---

# Additional files

## Supplementary files

• MDAR checklist

## Data availability

The serial blockface electron microscope volume is uploaded to BossDB (bossdb.org). The modeling code is publicly available on GitHub (https://github.com/pbmanis/vcnmodel (copy archived at *Manis, 2023a*) and https://github.com/cnmodel).The main simulation result files used to generate the figures in this manuscript have been uploaded to Dryad, and can be accessed at https://doi.org/10.5061/dryad.4j0zpc8g1. Simulation figures and figure panels can be generated using the DataTables script in the VCNModel package after downloading the simulation result files. All simulations shown in the paper, and/or their analyses, are included in the Dryad repository. They can be regenerated from the VCNModel package (above, on GitHub) using supplied scripts. Code and data for Figure 2—figure supplement 1 is in the file Figure2_Suppl1.py in the VCNModel GitHub repository.Code and data for Figure 5—figure supplement 2 is in pattern_summary.py in the VCNModel GitHub repository. Figures 1E, F, 2C, D, 3C,D,F,G, 7H and K, 8H were generated using Matlab code. The tables (Excel) and Matlab code are at https://github.com/gaspirou/pub_file_share (copy archived at *Spirou, 2023*).

*Continued on next page*

The following datasets were generated:

| Author(s) | Year | Dataset title | Dataset URL | Database and Identifier |
|---|---|---|---|---|
| Manis PB | 2023 | Data from: High-resolution volumetric imaging constrains compartmental models to explore synaptic integration and temporal processing by cochlear nucleus globular bushy cells | https://dx.doi.org/10.5061/dryad.4j0zpc8g1 | Dryad Digital Repository, 10.5061/dryad.4j0zpc8g1 |
| Spirou GA, Ellisman MH, Manis PB | 2023 | Serial blockface scanning electron microscopy of mouse cochlear nucleus nerve root region and compartmental computational models of reconstructed neurons | https://bossdb.org/project/spirou_manis2023 | BossDB, spirou_manis2023 |

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
