## [Editor Report]

This manuscript provides a structural analysis of bushy cells in the mouse cochlear nucleus. These neurons receive a large synaptic contact from the auditory nerve termed an endbulb that preserves the temporal information present in the auditory nerve, and are key elements of binaural sound localization circuits. The analysis combines volume electron microscopy techniques with computational models to predict heterogeneous bushy cell responses. The analysis takes morphological analysis of bushy cells to a new level, and the modeling is well done.

---

## [Decision Letter]

**Decision letter after peer review:**

Thank you for submitting your article "Two synaptic convergence motifs define functional roles for inputs to cochlear nucleus bushy cells" for consideration by *eLife*. Your article has been reviewed by 4 peer reviewers, including Catherine Emily Carr as the Reviewing Editor and Reviewer #1, and the evaluation has been overseen by Barbara Shinn-Cunningham as the Senior Editor. The following individual involved in the review of your submission has agreed to reveal their identity: Ian D Forsythe (Reviewer #2).

Essential revisions:

1. Rewrite to make the questions/hypothesis more relevant to the reader. Re-write abstract. Include more detail on the limitations of data and interpretation. Re-organise the whole text so that it reads in a more coherent fashion (bringing your results into greater clarity).

2. Add a discussion of different species and justify your choice of the mouse.

3. Revise the discussion to be more self-critical, particularly of your assumptions in the modelling, and limitations of integrating the structure and electrophysiology.

*Reviewer #1 (Recommendations for the authors):*

Your structural analysis of the bushy cells in mouse is outstanding, and the computational modeling is also well done.

Specific comments to be addressed, organized by section, start with "Application of the model" follow.

The statement that endbulb size does not strictly predict synaptic efficacy should be modified from an assertion to a prediction. It is possible that the cellular organization of the auditory nerve root region of the mouse cochlear nucleus represents two populations based on cells with smaller and larger somatic surface areas, but equally likely that they represent a continuum.

The synaptic convergence motifs of bushy cells suggest that there are two populations of BCs based on the absence or presence of high-efficiency suprathreshold inputs. Again, this is a hypothesis. Data, especially from Rothman and Manis, also support a continuum.

The section, mixed-mode cells operate in first-arrival and coincidence-detection modes when all inputs are active, evokes the reasonable concern that there may not be an obvious or linear relationship between synaptic efficacy/current and the physical size of the terminal.

The contribution of soma and dendrite area to bushy cell excitability suggests that soma surface area is a lesser determinant of excitability, and was weakly correlated with dendrite surface area. Again, the statement that the threshold is also only weakly correlated with soma area or the ratio of dendrite to soma areas requires electrophysiological validation and should be treated as a prediction.

The statement that the temporal precision of BCs varies by patterns of ASA should be converted to a prediction. It is likely that mouse bushy cell temporal precision of BC spiking exceeds that of ANFs, but this has not been shown for mouse.

The discussion is well written, but the statement that the convergence of weak and strong inputs regulates temporal fidelity should be changed to a prediction.

*Reviewer #2 (Recommendations for the authors):*

This reviewer is impressed by the ambition of the authors in conducting this work and appreciates the extremely impressive technical achievements. The authors should not be too defensive in their writing and should make more explanations of the limitations which will help this field move forward more rapidly.

It is very hard to work out what this manuscript is about from reading the abstract. The title of the manuscript does not really give a clear idea of the breadth of this paper. Both need to be rewritten and improved.

In addition, it is important during the rewrite to make the scientific narrative more easily understood by a casual reader. One way to do this would be to make each subtitle and the first sentence of each figure legend to be the take-home message of that figure.

There is an impressive amount of supplementary material, some of which is useful but I wonder whether all of it is necessary.

Ln 68-89: The last paragraph of the introduction is long and hard to understand until you've read the rest of the manuscript, so cut it and just provide a brief conclusion.

Ln 91 Results: The sub-headings are hard to understand and do little to guide the reader; I would advise making them more direct. E.g. Ln 153 "Number of convergent auditory nerve endbulb inputs exceeds previous estimates" > "5-12 endbulbs converge on each bushy neuron".

Ln 124: The synaptic convergence motifs of bushy cells. This is a definitive data set, it is fascinating to see the full panoply of synaptic structures in such a defined manner, and could be the subject of a complete manuscript on its own. Did I miss it, but where is the number of release sites per synapse? This would be important information and the manuscript feels incomplete without some attempt to show the number of release sites per endbulb. There is some potential confusion here in the use of 'synaptic density' synapses per unit area; and postsynaptic density that might define a release site… I realise that in line 212 you have a discussion of the application of the model and state that size does not predict efficacy, but the rationale for this statement may not be obvious to some readers. In many places in this manuscript, the caveats are left undisclosed or poorly explained. I think there is considerable scientific value in explaining the limitations of your interpretation.

Ln 172: Section Translating high-resolution neuron segmentation into compartmental models. To do this properly you should ideally have reconstructed labelled neurons from which the actual currents or APs had been measured. While I appreciate that you are developing a 'pipe-line' for dealing with many neurons, there seems little point in 'guessing' the neuronal conductances, when you've put so much effort into measuring precise structures…? Could you consider using your modelling to set limits on physiology?

Ln 249 and forward: A key problem is that the precise input-output characteristics for these particular neurons are unknown and the balance between the voltage and agonist-gated conductances is a significant unknown which impacts your interpretation. There is no way to check that you have the right balance of intrinsic conductance for the particular synaptic input. This lack of data severely undermines the security of your conclusions and needs to be carefully explained.

Ln 298: Your conclusion (mixed mode; latency or coincidence detection) seems a bit weak. Do you think that the mix-mode operation reflects a fundamental physiological phenomenon or a limitation to your ability to model the precise conductances in some of the neurons?

Ln 315: AIS length. This is interesting but has been studied in much more detail elsewhere – with the addition of physiological data, as you note in the discussion. The more novel observation here is the extent of endbulbs with contact with the AIS… This must have important consequences.

ln 377: BC dendrites exhibit novel branching patterns and structural features. This is a pedantic point, but the branching pattern is NOT novel, it is our recognition of it that is 'novel'. Perhaps you could say previously unrecognised….. It is an interesting observation. These dendritic expansions could be important, but their size begs the question as to what special conductances might accompany this structure, for which you have no evidence. However, the mitochondria data implies high metabolic activity… so there is much to investigate in future work.

ln 467: This complete map of the synaptic inputs onto these neurons is exciting and neat, but the result is diluted somewhat by not knowing which are inhibitory! This needs to be discussed. It is also interesting to find dendrites with no synaptic inputs…..But your assumption that these share the same properties as the other synapse-lined dendrites seem premature….. perhaps these naked dendrites have very different conductances?

The discussion should also include suggestions for what new data/experiments/methods would resolve ambiguities raised in this manuscript.

*Reviewer #3 (Recommendations for the authors):*

1. There are several apparent errors that need to be corrected: (1) Figure 2 – supplementary figure 2 and Figure 2 – supplementary figure 3 are identical. (2) Figure 4E-I: BC18, which has the largest ASA > 250um2, is not labeled as Mixed but Coincidence Detection group. Likewise, BC13 should be in the Coincidence Detection group but not in the Mixed group.

2. Despite the superb quality of 3-D reconstruction and detailed efforts in morphology-constrained modeling, the paper reads like a mass collection of data, without a clear question/hypothesis to be answered/tested. For example, the title of the paper is about "two input motifs' in the AN-bushy cell circuit; however, if we exclude the figures on methodology (Figures1 and 3), only Figures2, 4, and 7 are related to this idea, whereas Figures5, 6, 8 deal with other aspects of the circuit (Figure 5: whether inputs originate from common fascicles; Figure 6 and 8: discovery of special dendritic structures). As a result, the reader can be confused and get lost as they read the paper. The authors may want to reorganize the paper so that it reads more coherently.

3. Figure 4B and related supplementary figures and texts (line 249): the authors should describe how their "reverse correlation" is performed because it is absent in the Methods section. Also, the term "presynaptic coincidence rate" in Figure 4B can have two very different meanings: it can refer to the coincidences across all presynaptic inputs or the coincidences between the presynaptic spike and the postsynaptic spikes. Given the analysis applied here (reverse correlation), I think the authors likely refer to the latter; however, it can be confused with the "coincidence" in "coincidence detection", which means detecting coincidences across "pre" synaptic inputs. This can lead to the counter-intuitive interpretation that the author-defined "coincidence detectors" (e.g. Figure 4B BC05) have a low coincidence rate in all their presynaptic inputs (line 256). I suggest the authors use a more unambiguous term for the y-axis here: e.g. post/pre coincidence ratio, or maybe presynaptic spike rate.

4. (Line 33) Spherical bushy cells also show enhanced phase-locking, not only globular cells.

5. (line 72) references needed: which studies show it's hard to distinguish between SBC and GBC in mice?

6. (line 71) The rest of the paper still uses GBCs in many places (lines 88, 140, 224, 582, 604, 677). In fact, because the ultrastructures of the BCs in this paper match those of GBCs as reported in other species (as the authors described in Discussion, lines 511- 514), I suggest the authors use GBCs throughout the text and cite some references in the Discussion for problems distinguishing between SBC and GBC in mouse.

7. (line 112) The authors may want to explain why they didn't count synapses in every terminal. What are the (technical) limitations?

8. (line 153) Number of convergent inputs exceeds previous estimates: The authors should emphasize the previous results are "physiological" estimates in mouse brain slices. In fact, one of the authors' papers in cat (Spirou et al. 2005) shows GBCs have more than 4-6 inputs (15 – 23 inputs), and more in line with the observations here.

9. (line 267 – 285, and Figure 4E) It's a bit long-winding, but my understanding is that the goal is to see how likely a spike can be generated if you exclude the largest, the 1st, and 2nd largest, and so on, inputs? I think if the authors phrase it this way it will be easier for the reader to grasp the idea of coincidence vs mixed mode of detection (e.g. like what the authors describe later in lines 439 – 441).

10. (Figure 7P) I guess the take-home message here is that the largest input actually is not required for (enhanced) phase-locking to Fmods. So what's the functional role of the largest input here then? Do they improve, for example, the rate MTF or the entrainment of the spikes? The large endbulb synapse is the signature of the circuitry so it would be surprising that the large ones are actually not contributing more to the enhanced temporal precision.

11. (line 304) electromorphology – this is not a common term. Please define it.

12. (line 454) The VS is lower … because of peak-splitting in ANF. Peak-splitting is more common at high SPLs in response to pure tones; there are to my knowledge no in vivo reports of peak-splitting to low modulation frequencies, so this seems not very physiological.

13. (line 555) … they may grow or prune dendritic branches sufficiently rapidly…: Is this pure speculation or there is evidence for it? If the latter, please cite a reference.

14. (line 987) Since the model contains the axon, why not detect spikes from the axon (e.g. AIS) instead?

*Reviewer #4 (Recommendations for the authors):*

The collective data in the manuscript is impressive. The authors are innovative and smart. It's just that blockface SEM is quite limited in its resolution for synaptic structure, especially without the additional aid of tracer and/or antibody labeling.

---

## [Author Response]

Essential revisions:1. Rewrite to make the questions/hypothesis more relevant to the reader. Re-write abstract. Include more detail on the limitations of data and interpretation. Re-organise the whole text so that it reads in a more coherent fashion (bringing your results into greater clarity).

We have rewritten the abstract after substantially reorganizing the text, as the readers will note by the extensive red lining of the resubmission. The structural data and modeling are now less intertwined and more clearly separated. We first present core structural data (Figures 1-3), then describe our pipeline from high-resolution structure into realistic compartmental models (Figure 4), then present our hypotheses as tested via model predictions for input activation and responses to sound (Figures 5-6). We finish by presenting a second and shorter cycle of structural data and modeling by describing dendrite structure and the resulting model predictions (Figures 7-8). Limitations of our approach are expanded, mentioned throughout the text, and highlighted in the final section of the discussion. The re-write has, we think, made the overall presentation clearer.

2. Add a discussion of different species and justify your choice of the mouse.

We have added this justification to the first paragraph of the Results. Briefly, we chose the mouse because of the well-characterized biophysical properties of its bushy cells, the small size of its cell groups so that larger circuits can be captured inside of a given image volume, and availability of genetic and other molecular tools to further dissect synaptic circuitry in ways that can be mapped onto the current image volume and future image volumes of the same territory.

3. Revise the discussion to be more self-critical, particularly of your assumptions in the modelling, and limitations of integrating the structure and electrophysiology.

The second section of the Discussion now includes a focused presentation of the limitations of our approach. We also point out opportunities and challenges for the future extension of this kind of work. Edits throughout the Results and Methods also address the assumptions more directly.

Reviewer #1 (Recommendations for the authors):Your structural analysis of the bushy cells in mouse is outstanding, and the computational modeling is also well done.Specific comments to be addressed, organized by section, start with "Application of the model" follow.The statement that endbulb size does not strictly predict synaptic efficacy should be modified from an assertion to a prediction. It is possible that the cellular organization of the auditory nerve root region of the mouse cochlear nucleus represents two populations based on cells with smaller and larger somatic surface areas, but equally likely that they represent a continuum.

These data are now presented as subheading “Prediction 1” under the section re-titled “Model Predictions”. In “Prediction 3”, we extend this investigation and show that dendrite surface area correlates with excitability for a defined input size. We concur that, although two populations of endbulb size profiles, based on size of the largest input, emerge from this initial set of 21 cells, it is more likely that a continuum will be found in both the structural data and the modeling framework when more cells are reconstructed. Interestingly, the size of the largest input did not correlate with soma surface area (Supplemental Figure 2-1A), likely because soma surface area is tightly distributed except for one outlier (Figure 1). Conclusions from the structural data distributions given the size of the data set is now considered in the second and third sections of the Discussion.

The synaptic convergence motifs of bushy cells suggest that there are two populations of BCs based on the absence or presence of high-efficiency suprathreshold inputs. Again, this is a hypothesis. Data, especially from Rothman and Manis, also support a continuum.

As indicated for comment #1, these results are now presented under the section titled “Model Predictions”, “Prediction 1”. Inspection of the anatomical data (Supplemental Figures 2-2, 2-3) reveals that, for most of the bushy cells, the largest endbulb is considerably larger than all other inputs. This size distribution diverges more than is expected from previous reports, but in hindsight may be suggested from single ANF recording and labeling in cat (Liberman 1991, Figure 13). Although this sample of all inputs to single bushy cells is the largest to-date (21 cells with fully reconstructed auditory nerve inputs), it is certainly possible that the distribution of largest inputs will be more continuous when a larger sample is collected. As indicated in the previous comment, this possibility is considered in the fourth section of the Discussion.

The data from Rothman and Manis (2003) indicate a continuum for the strength of the low voltage activated conductance across CN cells. Note that these data were collected from acutely isolated neurons, so they report characteristics of the cell body membrane because most dendrite membrane was removed by the isolation process. These data suggest yet another parameter that can vary excitability, in addition to soma surface area. An experimental effort to measure both EPSC amplitudes (or better, conductance) and the density of the K^+^ conductance in bushy cells would be technically challenging because these two measures optimally require using very different internal electrode solutions (Cs-based for EPSCs, K-based for K currents). We address the potential variation in channel density among cellular compartments in the Discussion, as part of broader presentation of the range of parameters that can affect excitability of bushy cells.

The section, mixed-mode cells operate in first-arrival and coincidence-detection modes when all inputs are active, evokes the reasonable concern that there may not be an obvious or linear relationship between synaptic efficacy/current and the physical size of the terminal.

The reviewer’s comment is of course correct. There are multiple non-linearities that come into play, including those associated with the spike generator itself as well as the other conductances in the postsynaptic cell. Based on our analysis of images, it is highly likely that, even if not linear, the relationship between terminal size and EPSC amplitude is monotonically increasing. One linearizing factor is that the density of active zones is nearly constant across terminal size. Assuming similar vesicle release probability across active zones, more active zones will lead to more vesicles released per action potential, and larger EPSCs. For a given cell, postsynaptic factors also likely apply equally to all inputs. Based on the reviewer’s comment, we now mention in the Discussion these are additional factors that could adjust synaptic weights. Determining the distribution of release probability across active zones in the endbulb and calyx of Held has been an experimental goal for some time and is highlighted by our structural measurements and models.

The contribution of soma and dendrite area to bushy cell excitability suggests that soma surface area is a lesser determinant of excitability, and was weakly correlated with dendrite surface area. Again, the statement that the threshold is also only weakly correlated with soma area or the ratio of dendrite to soma areas requires electrophysiological validation and should be treated as a prediction.

As indicated for previous points, this suggestion has been addressed by creating a new section titled “Model Predictions”, “Prediction 3”.

The statement that the temporal precision of BCs varies by patterns of ASA should be converted to a prediction. It is likely that mouse bushy cell temporal precision of BC spiking exceeds that of ANFs, but this has not been shown for mouse.

This point is addressed also by presentation in the “Model Predictions” section, “Prediction 5”.

The discussion is well written, but the statement that the convergence of weak and strong inputs regulates temporal fidelity should be changed to a prediction.

Our intention is that section 2 of the Discussion, emphasis of model data as predictions in the Results, and the conclusion for the Discussion section mentioned here (“Convergence of weak and strong inputs regulates temporal fidelity”) will make it clear that we are presenting model data as hypotheses. The hypothesis that convergence regulates temporal fidelity in bushy cells has, as the reviewer knows, been part of the literature for several decades. We provide in this report the first detailed profile of all endbulbs onto individual globular bushy cells (21 cells, actually), and thus were able to incorporate biologically-inspired distributions of synaptic strengths for each cell. These data permit the predictions of experimental outcomes to be more precise, as hopefully emerges throughout the manuscript but especially in the section “Model Predictions”.

Reviewer #2 (Recommendations for the authors):This reviewer is impressed by the ambition of the authors in conducting this work and appreciates the extremely impressive technical achievements. The authors should not be too defensive in their writing and should make more explanations of the limitations which will help this field move forward more rapidly.

Please refer also to responses to public critique, #1-3. Essentially, our goal in the presentation is to convey that we collected, at great effort, the high-resolution structure for a purpose, which was to build a first-attempt compartmental model that was constrained by cell geometry and by known biophysical and physiological parameters. This approach to modeling serves to instantiate predictions from structure to function in an interpretable format, and exposes important missing information, the latter of which should lead to new experiments. The missing information, such as density of conductances across compartments, mapping of spontaneous rate onto particular endbulb morphologies, knowledge of release probability across active zones within and across endbulbs, and considerations of the sample size relative to the entire bushy cell population, are mentioned throughout the manuscript and are also collected in the second section of the Discussion, as mentioned in point #2 above. By additionally driving the model with auditory nerve activity profiles (generated using a published cochlear model), we make additional predictions of globular bushy cell sound-evoked activity that should also be tested experimentally. Hopefully, in the process of extensive rewriting, we communicate these messages in a less defensive manner. We thank the reviewer (and the other reviewers) for the comments that led to these changes.

It is very hard to work out what this manuscript is about from reading the abstract. The title of the manuscript does not really give a clear idea of the breadth of this paper. Both need to be rewritten and improved.

The title has been changed and the abstract was re-written. The abstract now hopefully conveys the central purpose and many details of the manuscript, as mentioned in the response to critique #1.

The new title is: “High-resolution volumetric imaging constrains compartmental models to explore synaptic integration and temporal processing by cochlear nucleus globular bushy cells”. We think this title indicates better the scope of the manuscript.

In addition, it is important during the rewrite to make the scientific narrative more easily understood by a casual reader. One way to do this would be to make each subtitle and the first sentence of each figure legend to be the take-home message of that figure.

We were not consistent in this style in the original submission and have made these edits throughout the manuscript. By collecting the model output into the sections “Model Predictions” with five subheadings, renaming other sections, and reorganizing the order of presentation (moving Figure 5 to Figure 3 to more clearly separate structural observations from the model), the flow should be more accessible because structure and model features and predictions are better separated and, we think, more easily linked.

There is an impressive amount of supplementary material, some of which is useful but I wonder whether all of it is necessary.

We have removed the supplementary material that is more methodological and not germane to major points of the manuscript (some of this can be presented in a separate manuscript). We also performed some reorganization of panels in the main and supplemental figures. These modifications were made:

Figure 2 Supplement 1: Because data points on the abscissa form a tight cluster, we zoomed in on the actual range of data values (no longer beginning at 0) for easier viewing. The axes for panel D were also swapped so that the abscissa matches panel C.Figure 2 Supplement 3: We now present the images for the mixed mode input profiles. Previously this figure was erroneously a repeat of the coincidence detection profiles (Supplemental figure 2); it now is the correct figure.

Figure 2 has 3 Supplemental figures, as in the original submission.

Due to reorganization, former Figure 3 is now Figure 4:

Former Figure 3 Supplement 4 (input impedances) was removed.

Current Figure 4 now has 4, instead of 5 Supplemental figures.

Due to reorganization, former Figure 4 is now Figure 5: Figure 5 Supplement 2 was deleted through these operations:

Panels B and D were moved to the main Figure 5

Supplement panel B became Main figure panel J

Main figure panel J became Main figure panel K, and was overlaid by the plot from Supplement 2, panel D.

The remaining panels are now reported in the text as numerical results.

Current Figure 5 now has 2, instead of 3 Supplemental figures.

Due to reorganization, former Figure 7 is now Figure 6: No changes were made to the Figure 6-Supplemental figures 1 and 4 (formerly 3).

Figure 6—Supplemental Figure 2 now has rate modulation transfer functions plotted as insets, to be consistent with Figure 6.

A new supplemental figure, Figure 6—Supplemental Figure 3, shows another measure of temporal firing, entrainment, for all 10 BCs, for 3 different input configurations and at two sound intensities. This figure is discussed in the Results. It echoes the rate modulation transfer functions and vector strengths, but provides another view (similar to the SAC data for clicks in Figure 6-supplemental figure 4)

Due to reorganization, former Figure 6 is now Figure 7: No changes were made to the 1 Supplemental figure for Figure 7.Supplemental figures to Main Figure 1 (1 Supplemental figure) and Main Figure 8 (no Supplemental figures) were not changed.

Ln 68-89: The last paragraph of the introduction is long and hard to understand until you've read the rest of the manuscript, so cut it and just provide a brief conclusion.

We have shortened this paragraph, which hopefully now makes the remaining information easier to digest.

Ln 91 Results: The sub-headings are hard to understand and do little to guide the reader; I would advise making them more direct. E.g. Ln 153 "Number of convergent auditory nerve endbulb inputs exceeds previous estimates" > "5-12 endbulbs converge on each bushy neuron".

We have changed this subheading and five others throughout the manuscript, added a sub-heading to the Discussion, and moved one sub-heading (Temporal precision of BCs varies by patterns of ASA) into a new sub-sub-heading named “Prediction 5: Temporal precision of GBCs varies by patterns of ASA”.

Ln 124: The synaptic convergence motifs of bushy cells. This is a definitive data set, it is fascinating to see the full panoply of synaptic structures in such a defined manner, and could be the subject of a complete manuscript on its own. Did I miss it, but where is the number of release sites per synapse? This would be important information and the manuscript feels incomplete without some attempt to show the number of release sites per endbulb. There is some potential confusion here in the use of 'synaptic density' synapses per unit area; and postsynaptic density that might define a release site…

We apologize for this oversight. This information was included in a previous version and was edited out and not replaced. We now provide the number of synapses per endbulb in a new Excel table “Dendrite_Quality_and_Surface_Areas_for_Sims_Cleaned_08_22_2022.xlsx” in the Dryad repository. We changed “synapse density” to “density of synapses”, which provides a value independent of the overall ASA for a particular endbulb. We use the ultrastructural definition of a synapse, which is comprised of an active zone, synaptic cleft, and postsynaptic density, and consider the nerve terminal to possess an ASA and contain many synapses.

I realise that in line 212 you have a discussion of the application of the model and state that size does not predict efficacy, but the rationale for this statement may not be obvious to some readers. In many places in this manuscript, the caveats are left undisclosed or poorly explained. I think there is considerable scientific value in explaining the limitations of your interpretation.

We now intersperse considerations of limitations of the methods more evenly throughout the Results, and also collect them into the second section of the Discussion, where we also provide the larger context for our approach. In response to this specific comment, we add a statement at the end of this section (Prediction 1 under Model Predictions) that refers the reader to Predictions 3 and 4, where other underlying mechanisms that affect excitability are explored.

Ln 172: Section Translating high-resolution neuron segmentation into compartmental models. To do this properly you should ideally have reconstructed labelled neurons from which the actual currents or APs had been measured. While I appreciate that you are developing a 'pipe-line' for dealing with many neurons, there seems little point in 'guessing' the neuronal conductances, when you've put so much effort into measuring precise structures…? Could you consider using your modelling to set limits on physiology?

The reviewer raises an interesting point, which exposes limitations in current experimental technology. While this combination of data will be useful, currently this approach is incompatible with high quality ultrastructure and volume EM. The recording process disrupts synaptic inputs and, if brain slices are used, the degradation of tissue integrity begins from the moment of slicing. Our data serve as a template for future recording and light microscopy studies that can be matched to the volume EM dataset, provided procedures are in place to also count and quantify endbulb inputs from the same cells. In light of the suggestion to use modeling to evaluate existing physiology, we did use recorded data from mice to constrain the ion channel types and densities in the model (see Methods, Ion Channels and Receptors, Conductance Scaling). The model does constrain the physiology, because we utilize these biophysical metrics, and the SWC/HOC representations for NEURON simulations were constructed at a very high spatial resolution, close to anatomical reconstructions from electron microscopy, which are as exact as current technology permits. We used values for conductances that are largely from the literature (without “tuning”). For example, the value chosen for K_LV_ in bushy cells was at approximately the median for recordings from cells closest to the nerve root, where our image volume was located (Cao et al. 2007). In future studies we can explore variations in conductance values in the model, recognizing that this is a very large parameter space. In addition, we discuss some of the experimental challenges (presented above) to collect all necessary data from the same cell.

Ln 249 and forward: A key problem is that the precise input-output characteristics for these particular neurons are unknown and the balance between the voltage and agonist-gated conductances is a significant unknown which impacts your interpretation. There is no way to check that you have the right balance of intrinsic conductance for the particular synaptic input. This lack of data severely undermines the security of your conclusions and needs to be carefully explained.

As indicated in the response to point 11, the models are constrained by robust experimental measurements in the same species and location within the nucleus. Hence, any variation in voltage-gated conductances would occur within the reported range of values, to the extent the reported experimental samples are representative of the globular bushy cell population. For a particular cell, the synaptic conductances of all endbulbs across the size range will engage with a shared set of postsynaptic ion channel conductances. The other important question raised by the reviewer is whether endbulbs differ in their weight based on simple scaling by the number of synapses they form with the postsynaptic cell. For this issue, too, we benefit from the use of mice as the experimental animal, since recordings of miniEPSCs have been made in this species. These recordings have not revealed subpopulations of bushy cells with differing mean values (despite what we now know to be different complements of endbulb sizes), so we conclude that parameters such as the number of postsynaptic receptors or synaptic vesicle volume, which could affect synaptic weight, vary similarly across endbulbs. We mention these topics in the new Discussion section “Toward a complete computational model for globular bushy cells; strengths and limitations” and think that our presentation now highlights these topics as both based on published literature yet requiring further investigation.

Ln 298: Your conclusion (mixed mode; latency or coincidence detection) seems a bit weak. Do you think that the mix-mode operation reflects a fundamental physiological phenomenon or a limitation to your ability to model the precise conductances in some of the neurons?

The mixed mode arises from consideration of the range of sizes of endbulbs, as well as an interaction with the postsynaptic conductances that set spike threshold, and is very likely a physiological property of these cells. A key contribution of our experimental data is to provide the size distribution of endbulb inputs, along with the number of synapses per terminal as an estimate of synaptic weight. The size of the largest inputs onto the mixed mode cells is so different from the sizes of the inputs onto coincidence detection cells that we are confident in our hypothesis, based on measurements of mEPSCs discussed in the previous point, that these terminals have a much higher probability of generating spikes. In addition, studies that have investigated the auditory nerve EPSCs in mouse bushy cells using a range of electrical stimulus strengths with electrodes in the auditory nerve root have sometimes reported the presence of weaker (500-1000 pA) inputs at low stimulus levels, suggestive of “smaller” end bulb inputs (Cao and Oertel, 2010, Figure 3C,D; and multiple similar observations from the Manis lab). As a caution regarding the interpretation of these results, with electrical stimulation it is not possible to be 100% confident that these are auditory nerve inputs, but it is likely. Notably, in the model, the larger inputs (> 180 µm^2^) have a high probability of generating spikes even in the cells that exhibit lower excitability in the models, supporting the notion that the large inputs indeed have greater weight. Hence, we propose that this topic is important and worthy of further investigation. If terminal size were inconsequential to synaptic weight, then one must wonder why some terminals are so much larger than others. The conclusion of two populations is based on the histogram in figure 2D; this distribution may reflect more of a continuum as more globular bushy cells are analyzed at high resolution, but the range of data will likely not change. As stated in a previous critique, experiments using optical manipulation and optical recording in a brain slice preparation would permit activation of each input individually and postsynaptic recording without damaging somatic inputs. Notably, even among globular bushy cells with a smaller range of input sizes, values vary by a factor of 4 (35 – ~140µm^2^), and we predict that the largest of these inputs contribute to generation of a large percentage of postsynaptic spikes. Thus, the size range of inputs, revealed here for the first time, affects firing in both the coincidence detection and mixed mode cells.

Ln 315: AIS length. This is interesting but has been studied in much more detail elsewhere – with the addition of physiological data, as you note in the discussion. The more novel observation here is the extent of endbulbs with contact with the AIS… This must have important consequences.

We agree that the functional relevance of our observation that endbulbs partially extend onto the AIS does require further investigation and constitutes a study of its own. The coupling of current from soma to the initial segment also requires further consideration because that factor will determine the relative efficacy of synaptic inputs onto the AIS. As we indicate in the Results (Prediction 3) and Discussion section “Multiple cellular mechanisms to tune excitability”, identifying neurochemical phenotypes of synaptic inputs to the AIS and acquiring better measures of ion channel density along the AIS length will permit more meaningful exploration of these parameters and their control of bushy cell activity. These synaptic inputs can be mapped onto the compartmental models as their cellular sources are identified. The exact metrics that we report on the variance in lengths, diameters and innervation of the AIS are, we think, important experimental advances in understanding overall integration of synaptic inputs.

ln 377: BC dendrites exhibit novel branching patterns and structural features. This is a pedantic point, but the branching pattern is NOT novel, it is our recognition of it that is 'novel'. Perhaps you could say previously unrecognised….. It is an interesting observation. These dendritic expansions could be important, but their size begs the question as to what special conductances might accompany this structure, for which you have no evidence. However, the mitochondria data implies high metabolic activity… so there is much to investigate in future work.

The viewer correctly points out our poor use of the word “novelty”. We meant to convey that this type of branching, often at obtuse angles, the presence of primary and often secondary hub locations in the dendritic tree, and the collection of branches into a bundle in which braiding can occur, are different from other neurons in the mammalian brain. We now describe in the Results the uniqueness of the structures and have changed the wording of the heading to that section of the manuscript.

The hubs are, indeed, intriguing structures. Do unique collections of conductances, pumps or adhesion molecules populate the hubs, in particular the main hub? One outcome of this work is to identify future topics for study and, of course, the investigation cannot commence until the thing to be investigated has been identified. We hazard a functional prediction for hubs in the Discussion (keep branching tight and electrotonic structure small) and agree that there are many related topics to pursue.

ln 467: This complete map of the synaptic inputs onto these neurons is exciting and neat, but the result is diluted somewhat by not knowing which are inhibitory! This needs to be discussed. It is also interesting to find dendrites with no synaptic inputs…..But your assumption that these share the same properties as the other synapse-lined dendrites seem premature….. perhaps these naked dendrites have very different conductances?

When highlighting limitations of the data in the second section of the Discussion, we state clearly that we do not know the neurochemical phenotype of the terminals. As we continue to analyze the image volume, we will be able to reconstruct dendritic inputs and their axons. As indicated in the Discussion, these data will offer clues to the origin of the fibers when matched with data collected in future EM volumes and from light microscopy experiments that label axons from known cell types and cell groups. Note that our imaging technologies are improving, and in future image volumes we anticipate being able to assess synaptic vesicle shape and to label subclasses of neurons for detection in EM. Those data, too, can then be mapped onto these existing high-resolution compartment models. Assessing conductances in naked dendrites emerges now as a topic of interest, and will require quantifiable, high-resolution techniques to find an answer. We have begun considering how to explore the hubs in particular, and the distribution of dendritic conductances in general.

The discussion should also include suggestions for what new data/experiments/methods would resolve ambiguities raised in this manuscript.

In our reorganization of the Discussion, we mention some of these approaches, and hopefully strike a balance between interpretation of the current project, suggesting new do-able experiments, and predicting future experiments for which, in many cases, the techniques do not yet exist. For example, certain model manipulations, such as removing selected dendrite branches or somatic inputs while maintaining cell integrity other aspects of neural circuit viability, are not yet in the realm of experimental capability. The ability to perform such manipulations, we think, is part of the value of high-resolution structure and compartmental models at this scale and level of detail.

Reviewer #3 (Recommendations for the authors):1. There are several apparent errors that need to be corrected:(1a) Figure 2 – supplementary figure 2 and Figure 2 – supplementary figure 3 are identical.

Our apology. The correct Figure 2 – Supplementary Figure 3 is now included in the manuscript.

(1b) Figure 4E-I: BC18, which has the largest ASA > 250um2, is not labeled as Mixed but Coincidence Detection group. Likewise, BC13 should be in the Coincidence Detection group but not in the Mixed group.

Thank-you for catching this error, which has now been corrected so that all groups are referred to correctly in figures and the text.

2. Despite the superb quality of 3-D reconstruction and detailed efforts in morphology-constrained modeling, the paper reads like a mass collection of data, without a clear question/hypothesis to be answered/tested. For example, the title of the paper is about "two input motifs' in the AN-bushy cell circuit; however, if we exclude the figures on methodology (Figures1 and 3), only Figures2, 4, and 7 are related to this idea, whereas Figures5, 6, 8 deal with other aspects of the circuit (Figure 5: whether inputs originate from common fascicles; Figure 6 and 8: discovery of special dendritic structures). As a result, the reader can be confused and get lost as they read the paper. The authors may want to reorganize the paper so that it reads more coherently.

This comment was made also by Reviewers 1 and 2. Please refer to those responses. In summary, we have extensively re-written and re-organized the presentation, including a change of title. We hope that these changes clearly convey the sequence from collection of experimental data (high-resolution structure), the pipeline for those structural data and published biophysical data into a compartmental modeling framework, and the generation of multiple functional hypotheses based on manipulations of model geometry and input of auditory nerve spike trains derived from a published cochlear model.

3. Figure 4B and related supplementary figures and texts (line 249): the authors should describe how their "reverse correlation" is performed because it is absent in the Methods section. Also, the term "presynaptic coincidence rate" in Figure 4B can have two very different meanings: it can refer to the coincidences across all presynaptic inputs or the coincidences between the presynaptic spike and the postsynaptic spikes. Given the analysis applied here (reverse correlation), I think the authors likely refer to the latter; however, it can be confused with the "coincidence" in "coincidence detection", which means detecting coincidences across "pre" synaptic inputs. This can lead to the counter-intuitive interpretation that the author-defined "coincidence detectors" (e.g. Figure 4B BC05) have a low coincidence rate in all their presynaptic inputs (line 256). I suggest the authors use a more unambiguous term for the y-axis here: e.g. post/pre coincidence ratio, or maybe presynaptic spike rate.

Thank you for this comment. We appreciate now that the terminology used is ambiguous. We have changed the wording to “Cross-correlation”, and added a section to the Methods (called “Cross- correlation”) describing the calculation method, referencing the original code from the package Brian, and clearly delineating what the “cross-correlation” represents. The window time that is plotted is only for ANF events in the time window prior to the postsynaptic spike. The label in Figure 5B (old Figure 4B) has been changed from “Presynaptic Coinc. Rate (Hz)” to “Pre-Post Coinc. Rate”

4. (Line 33) Spherical bushy cells also show enhanced phase-locking, not only globular cells.

Our intention was to convey that both SBCs and GBCs exhibit enhanced phase-locking, but that this phenomenon has greater incidence in GBCs (Joris et al. 1994a, Figure 4; Joris et al. 1994b, Figure 5). We now use that more explicit language in that sentence.

5. (line 72) references needed: which studies show it's hard to distinguish between SBC and GBC in mice?

In the process of simplifying the Introduction, we have moved this point to the first paragraph of the Discussion. There, we review the classification criteria, which mostly align with criteria for GBCs used in cat. Lauer et al. (2013) preferred a distinction of BC subtypes in mouse based on the number of auditory nerve inputs and not cytological criteria, as we now mention in the Discussion. By their criteria these cells would also qualify as GBCs.

6. (line 71) The rest of the paper still uses GBCs in many places (lines 88, 140, 224, 582, 604, 677). In fact, because the ultrastructures of the BCs in this paper match those of GBCs as reported in other species (as the authors described in Discussion, lines 511- 514), I suggest the authors use GBCs throughout the text and cite some references in the Discussion for problems distinguishing between SBC and GBC in mouse.

Since volume EM provides a new view of bushy cells (previous ultrastructural characterizations in the literature were based on single sections or a few serial sections), we prefer to begin with the generic term “bushy cell”. To introduce the classification of these cells as GBCs, we added these sentences to the second paragraph of the results after the description of the cytological features of the cells: “Based on these cytological criteria, location of cells in the auditory nerve root, and multiple endbulb inputs (see below), we classify these cells as globular bushy cells (GBC). We use that notation throughout the remainder of the manuscript.” We then start a new paragraph to discuss the somatic synaptic configuration.

7. (line 112) The authors may want to explain why they didn't count synapses in every terminal. What are the (technical) limitations?

Since this effort is performed manually, the technical limitation is only time. We identified 180 endbulb terminals onto 21 BCs in the image volume. After counting synapses in 11% of the terminals and analyzing their distribution, we determined that further effort on this activity would not yield additional meaningful information relative to the many other segmentation tasks that were performed. We are currently implementing machine learning methods to speed up automated synapse detection in this and future image volumes, but that effort is not yet sufficiently refined.

8. (line 153) Number of convergent inputs exceeds previous estimates: The authors should emphasize the previous results are "physiological" estimates in mouse brain slices. In fact, one of the authors' papers in cat (Spirou et al. 2005) shows GBCs have more than 4-6 inputs (15 – 23 inputs), and more in line with the observations here.

We changed the heading for this section (now reads “Five-12 auditory nerve endbulbs converge onto each globular bushy cell”), in accord with the suggestion by Reviewer #2 to provide more descriptive subheading titles. The comparison with physiological measures in mouse is noted in this paragraph. In the first paragraph of the Discussion, we mention the correspondence with the large number of inputs in the cat, as reported by Spirou et al. (2005).

9. (line 267 – 285, and Figure 4E) It's a bit long-winding, but my understanding is that the goal is to see how likely a spike can be generated if you exclude the largest, the 1st, and 2nd largest, and so on, inputs? I think if the authors phrase it this way it will be easier for the reader to grasp the idea of coincidence vs mixed mode of detection (e.g. like what the authors describe later in lines 439 – 441).

We have revised the text in that sub-section (“Mixed-mode cells operate in both latency and coincidence-detection modes when all inputs are active”), which is part of the newly named section “Model Predictions”. The specific text was moved to the “Model Predictions” section as subheading, “Prediction 5: Temporal precision of globular bushy cells varies by distribution of endbulb size” to make the entire presentation easier to follow, as the reviewer suggested.

10. (Figure 7P) I guess the take-home message here is that the largest input actually is not required for (enhanced) phase-locking to Fmods. So what's the functional role of the largest input here then? Do they improve, for example, the rate MTF or the entrainment of the spikes? The large endbulb synapse is the signature of the circuitry so it would be surprising that the large ones are actually not contributing more to the enhanced temporal precision.

The reviewer makes an insightful observation. One surprise in this data set is that the largest endbulb onto nearly one-half of GBCs (9/21) is likely suprathreshold (based on structural ­analysis and model predictions). As suggested by the reviewer, we conducted additional analyses to calculate the rate (r)MTF and entrainment. The suprathreshold input alone leads to a lower rMTF in the GBC than ANFs at all modulation frequencies, but in combination with the subthreshold inputs, generates a higher rate for modulation frequencies ≥ 200 Hz. The subthreshold inputs alone, however, did not drive the rMTF above that of the AN in any case. These model results are presented as insets to panels in Figure 6O1-4 and Supplemental Figure 6-2A-F. Similarly, the largest input, when added to the subthreshold inputs, increased the entrainment index for SAM at 300 and 400 Hz. These analyses are presented in a new Supplemental Figure to Figure 6, Supplement 3. We added text to the section Model Prediction 5 to present these data.

11. (line 304) electromorphology – this is not a common term. Please define it.

We replaced this term with the more common “electrotonic structure”, and added a brief explanation.

12. (line 454) The VS is lower … because of peak-splitting in ANF. Peak-splitting is more common at high SPLs in response to pure tones; there are to my knowledge no in vivo reports of peak-splitting to low modulation frequencies, so this seems not very physiological.

Thank you for this observation. We agree that “peak-splitting” was not the correct terminology. At low modulation frequencies, the duration of each modulation cycle is above threshold long enough that several spikes can be generated. This spike pattern degrades spike timing within each modulation cycle and results in a lower vector strength value. We now make this observation in the new paragraph describing rMTF simulations.

13. (line 555) … they may grow or prune dendritic branches sufficiently rapidly…: Is this pure speculation or there is evidence for it? If the latter, please cite a reference.

Dendrite remodeling, defined as loss or gain of dendrite branches and length, in adult animals is well-documented in response to physical injury, emotional stress (PTSD), and disease states (for review, see Furusawa and Emoto, Front. Cell Neurosci, 2021). Experience-dependent remodeling has also been described for inhibitory neurons in cortex (for review Berry and Nedivi, Ann Rev Vis Sci, 2016) and in response to LTP and LTD paradigms (Monfils et al. 2000; Monfils & Teskey 2004). Remodeling of dendrite structures has been tied to cyclic physiological patterns, such as diurnal or estrous cycles (ventral hypothalamus neurons; Griffin et al. 2010; Ferri & Flanagan-Cato 2012). These studies did not assess the speed of dendrite growth or pruning, although dramatic increases in dendrite length have occurred at 2 hours following emergence from hibernation (shortest interval tested in von der Ohe et al. 2006). We summarize these studies briefly in the Discussion section “Multiple cellular mechanisms to tune excitability”, to add context to our hypothesis.

14. (line 987) Since the model contains the axon, why not detect spikes from the axon (e.g. AIS) instead?

The model permits recording of activity from any compartment. Adding compartments increases the amount of data which must be stored and analyzed. We employed somatic recordings to be congruous with most experimental data. We implemented spike identification procedures (Hight and Kalluri, 2016) and after proofreading the waveforms we are confident in the quality of the spike data. Based on our first report here on axon initial segment (AIS) geometry and innervation, we plan to collect more data on AIS features, myelin thickness, and distance to first node (it may be short), and then compare recording sites for future modeling efforts.

Reviewer #4 (Recommendations for the authors):The collective data in the manuscript is impressive. The authors are innovative and smart. It's just that blockface SEM is quite limited in its resolution for synaptic structure, especially without the additional aid of tracer and/or antibody labeling.

We interpret this comment to reflect the inability to map neurochemical phenotype and cellular source to many of the reconstructed inputs. As indicated in responses also to the other reviewers, the image resolution is a tradeoff between time and volume. We selected this resolution to permit synapse identification and acquire a large number of neurons at nanoscale resolution. We have also created a spatially detailed compartmental model which provides a framework onto which inputs from identified source neurons can be mapped once those experiments are completed. This activity entails future work with the current image volume in which small inputs onto all BCs, and their linking axons, are traced. Additionally, it requires future experiments, likely requiring viral vector labeling, whereby individual neuron sources are identified and typed, their axons are fully reconstructed, in some cases across the entire brain (e.g., descending projections from auditory cortex to the CN), and the resulting branching patterns can be associated with those reconstructed from this image volume and other SBEM volumes that we collect. Note that most of the existing set of tracer experiments, whereby brain regions are injected with an anterograde tracer and axons mapped in a target region, are not sufficient for the kind of cell type to cell type connectome that we are attempting to build and model. To complement viral vector injections, genetic marking of cells using an electron dense label may become possible at sufficient resolution once specific cell type markers and the appropriate Cre lines have been made. The future is challenging and bright, and we think that our methods create a template onto which additional information can be mapped.